# Sensitivity of the Regional Climate Model MAR's snowpack to the assimilation parametrization of satellite-derived wet-snow masks on the Antarctic Peninsula

Thomas Dethinne[1,2], Quentin Glaude[1,3], Ghislain Picard[4], Christoph Kittel[5], Anne Orban[3], and Xavier Fettweis[1]

[1]University of Liège, SPHERES research unit, Laboratory of Climatology, Liège, Belgium
[2]University of Liège, SPHERES research unit, Earth Observation and Ecosystem Monitoring laboratory, Liège, Belgium
[3]University of Liège, Centre Spatial de Liège, Signal Processing Laboratory, Liège, Belgium
[4]Institut des Géosciences de l'Environnement (IGE), Université Grenoble Alpes/CNRS/UMR 5001, Grenoble, France
[5]Institut des Géosciences de l'Environnement (IGE), Université Grenoble Alpes/CNRS/IRD/G-INP, Grenoble, France

**Correspondence:** Thomas Dethinne (tdethinne@uliege.be)

**Abstract.** This paper discusses the use of regional climate models (RCMs) and remote sensing (RS) data to study climate change in remote regions such as the polar regions. RCMs can simulate how certain climate variables, such as surface melt, runoff, and snowfall, are likely to change in response to different climate scenarios, but they are subject to biases and errors. RS data can assist in reducing and quantifying the uncertainties of the model by providing indirect observations of the modeled variables upon the present climate. In this work, we investigate the sensitivity of the RCM "Modèle Atmosphérique Régional" (MAR) to the parameters of the assimilation of wet snow occurrence estimated by RS datasets. The assimilation is performed by nudging the MAR snowpack temperature to match the presence of liquid water observed by satellite. The sensitivity is tested by modifying parameters of the assimilation, such as the depth to which the MAR snowpack is warmed up or cooled down to match with the satellite based wet-snow extent, the quantity of water required into the snowpack to qualify a MAR pixel as wet or not (0.1 or 0.2% of the snowpack mass being water), and assimilating different RS datasets. The data assimilation is performed over the Antarctic Peninsula over the 2019-2021 period. The results show an increase in surface melt (+66.7% on average, or +95 Gt) going along with a small decrease in surface mass balance (SMB) (-4.5% on average, or -20 Gt) for the 2019-2020 melt season. The model is sensitive to the tested parameters, albeit with varying orders of magnitude. The assimilation depth has more impact on the resulting surface melt than the quantity of liquid water content (LWC) required in the snowpack due to strong refreeze occurring in the top layers of the snowpack. The values tested for the quantity of LWC required into the snowpack to qualify a MAR pixel as wet or not are lower than during typical melt days (approximately 1.2%) and impact mainly at the beginning and the end of the melting period. The assimilation will allow an uncertainty estimation of MAR melt production and identify potential issues in the near-surface snowpack modeled processes. This paves the way for improving models to achieve more accurate simulations of the future.

## 1 Introduction

More than two-thirds of the Earth's freshwater is held in the polar ice sheets (Church et al., 2013), with the majority of it trapped as ice on the ground at the south pole, forming the Antarctic Ice Sheet (AIS). According to Fretwell et al. (2013), if all the ice in the AIS was to melt, it would result in a sea-level rise of 56 meters. Currently, the AIS is primarily losing mass due to grounded ice flowing into the ocean. There, the ice is lost mainly through a combination of basal melting and calving (The IMBIE Team, 2018; Rignot et al., 2019; Adusumilli et al., 2020).

However, the surface melt production on the ice sheet is important for several reasons. Even moderate surface melt over the ice shelves, the floating boundaries of the ice sheet, is thought to weaken the shelf structure and to cause ponding and hydrofracturing, leading to substantial mass loss (Scambos et al., 2003; Lai et al., 2020) and, surface melting is becoming a growing concern as it is taught to increase greatly with climate change (Trusel et al., 2015; Bell et al., 2018; Gilbert and Kittel, 2021). Ice shelves exert a buttressing effect on the upstream ice flow, regulating the amount of ice that reaches the surrounding ocean. As they thin from mass loss, this buttressing effect is reduced (Favier and Pattyn, 2015; Paolo et al., 2015), and AIS ice flow velocity is increased.

Climate models are nowadays one of the handiest tools to study polar climate evolution. Some of them also include the possibility to model the evolution of the snowpack. A notable example is MAR (for "Modèle Atmosphérique Régional" in French), a Regional Climate Model (RCM) especially developed to monitor the polar climate and the surface mass balance of both ice sheets.

Proper modeling of the surface melt is required to study both the conditions leading to the destabilization as hydrofracturing is impacted by the melting/snowfall ratio and by the snowpack capacity to retain and refreeze meltwater (Donat-Magnin et al., 2021; Gilbert and Kittel, 2021), but also to study the evolution of the snowpack during strong melt events. Studying the snowpack ability to retain liquid water is crucial because the Antarctic snowpack could saturate, and stop absorbing surface meltwater in the future, as it is modeled currently over the Greenland ice sheet (Noël et al., 2017).

However, RCMs still have some limitations. Because of the forcing, or the physical assumptions, the models may contain significant uncertainties. These uncertainties can be mitigated by employing external data, which is not already incorporated into the model, to improve its accuracy at specific points in space and/or time. This technique is known as "data assimilation" and is commonly applied in numerous fields where observations can be integrated into a model (Evensen, 2009; Navari et al., 2018).

Assimilation of data into the model is a crucial step in quantifying the uncertainties associated with the model output without assimilation. The assimilation process helps to identify areas and periods where the simulations are not consistent with the observations. This can help us better understand the underlying physical processes and their interactions. Accordingly, data assimilation provides a powerful tool for improving the reliability of models. In our case, it is an essential step in the process of model refinement, leading to improved predictions of future scenarios.

The highly uneven topography of the area is challenging for the RCMs usually operating at a 10-kilometer spatial resolution. Phenomena depending on very local conditions such as melt induced by the Foehn effect can occur at a smaller spatial scale

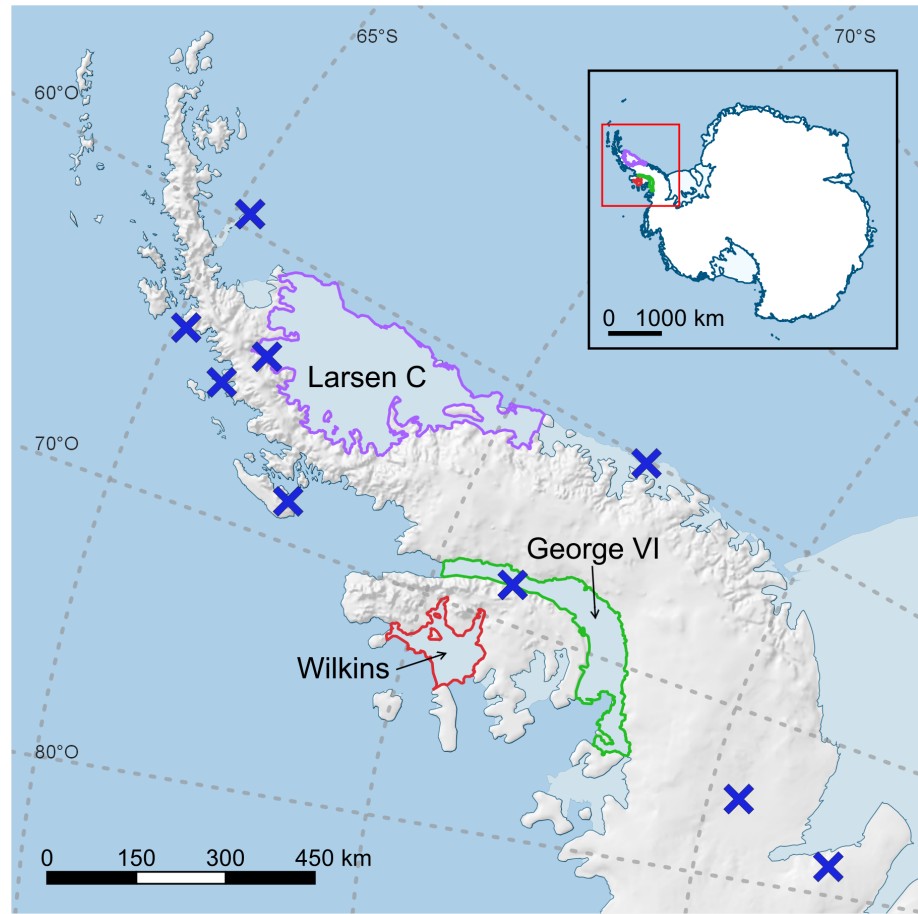

**Figure 1.** Locations of the Antarctic Peninsula and the three studied ice shelves. The ice shelves are denoted by color outlines. Larsen C is outlined in purple, George VI in green, and Wilkins in red. Blue crosses indicate the position of the weather stations used for the model's evaluation (Sect. 3).

than the spatial resolution of RCMs and thus may be mitigated by the model (Datta et al., 2019; Chuter et al., 2022; Wille et al.,
2022). However, high-resolution satellites can document these local events that could be missed by RMCs in case of localized or extreme events.

In this paper, we assimilate satellite-derived binary wet-snow masks (wet/non-wet) over the Antarctic Peninsula (AP), West Antarctica, into the MAR model for two melt seasons (2019-2020 and 2020-2021). Three major ice shelves are located over the AP: Larsen C, George VI, and Wilkins (Figure 1). These ice shelves undergo the most surface melt of the AIS, and their surface
processes are also poorly understood, with complex surface hydrology (Barrand et al., 2013; Datta et al., 2018; Johnson et al., 2020). Presently, assimilating remotely-sensed products in RCMs is a promising method to quantify the surface meltwater quantity in Antarctica. The scarcity of field observations and the complexity of the surface hydrology (Bell et al., 2018) make it difficult to evaluate and constrain models otherwise.

The assimilation algorithm performed in this paper is derived from the framework described in Kittel et al. (2022) where MAR near-surface snowpack is warmed up or cooled down to better match satellite-derived wet-snow masks. In this study, experiments have been performed by varying the depth to which the snowpack temperature is changed (called the assimilation depth hereafter) to match satellites, the minimum liquid water quantity to consider the modeled snowpack state as wet, and the assimilated wet-snow product to test the sensitivity of the model to the assimilation.

The satellite data, the model, and the assimilation are presented in Sect. 2. The validation of the model is described in Sect. 3. The results of the sensitivity tests when assimilating data into the model are discussed in Sect. 4. Finally, a general conclusion and discussions on the perspectives of the assimilation of remote sensing data in the MAR model are included in Sect. 5.

## 2 Methods and data

### 2.1 Satellite data

Depending on the context of the study, like the region of interest, the length of the simulation, or the spatial resolution, the use of one specific satellite dataset over another for the assimilation can be useful. Reckoning on the sensor and acquisition times, wet-snow occurrences derived from satellites can differ (Husman et al., 2022). Some sensors tend to have coarser resolution and provide information with higher uncertainties in areas with complex topography but provide longer time series of daily images with wet snow detection algorithms that have proven to be efficient (Zwally and Fiegles, 1994; Colosio et al., 2021). On the other hand, other sensors have a better spatial resolution but may have a lower revisit time. The choice of the satellite dataset can thus influence the results of the assimilated model.

We employed three satellite datasets (Table 1) to create the binary (dry/wet) wet-snow masks assimilated. The three datasets are derived from sensors operating in the microwave spectrum (in the GHz frequencies). Among them, one is called a "passive sensor" meaning the sensor records Earth's natural radiations, while the other two are classified as "active sensors" since they actively emit electromagnetic pulses to illuminate the area covered by the satellite. Microwave operating sensors are commonly used to map snow cover, sea ice, or the extension of wet snow over ice sheets (Parkinson, 2001; Colosio et al., 2021). The signal is used to detect if the snowpack is wet as microwaves interact with water. The presence of liquid water in the snowpack induces a change in its emissivity and absorptivity. This change leads to a change in the satellite measurements, the backscattering coefficient $\sigma_0$ for active sensors and the brightness temperature for passive sensors (Zwally and Fiegles, 1994; Johnson et al., 2022; Picard et al., 2022). In this study, the presence of wet snow detected by satellites is interpreted as the presence of liquid water underneath or at the surface of the snowpack. Using microwaves also brings other advantages such as atmospheric transparency and day-and-night acquisitions. However, the lower spatial resolution of passive microwave sensors (generally 10 to 50 km) compared to active sensors (generally 10 m to 5 km) is problematic to determine small-scale melt extents (Datta et al., 2018). Finally, with pixels of 100 km2 (for AMSR2 - See Table 1), a majority of the pixels are overlapping regions with different land cover or surface height (Johnson et al., 2020).

**Table 1.** Technical specifications of the remote sensing datasets employed for the assimilation. Datasets are referred to by the name in bold characters in the paper

| Plateform | Sensor | Sensor type | Pixel size | Frequency (GHz) | Revisit time (days) | Reference |
|---|---|---|---|---|---|---|
| **Sentinel-1 (S1)** | C-SAR | Active | 10-40m | 5.405 | 6 | ESA (2023) |
| Metop | **ASCAT** | Active | 4.45km | 5.255 | 1 | EUMETSAT (2023) |
| GCOM-W1 | **AMSR2** | Passive | 10km | 18.7 | 2 | JAXA (2021) |

### 2.1.1 Advanced Microwave Scanning Radiometer 2

In this study, we used the Advanced Microwave Scanning Radiometer 2 (AMSR2) aboard the Global Change Observation Mission - Water "SHIZUKU" (GCOM-W1) retrieved from the Japan Aerospace Exploration Agency (JAXA) G-Portal (JAXA, 2021). Thanks to the sun-synchronous orbit at an altitude of 700 $km$ and large swath, low-resolution daily observations of the polar regions are obtained. We used the level-3 products containing the daily mean brightness temperature in horizontal polarization in the 18.7 GHz channel, resampled at a 10 $km$ resolution. The 18.7 GHz channel is used as it is slightly more sensitive to liquid water content than the other frequencies (Picard et al., 2022). Ascending (satellite path goes from south to north) and descending (satellite path goes from north to south) paths were processed separately, as they respectively happen in the morning and in the evening. The separated processing allows the creation of two wet-snow masks from one dataset. Wet-snow detection with AMSR2 is based on a change in the snowpack physical properties. A dry snowpack has a lower emissivity ($\epsilon$) than a wet snowpack (Zwally and Fiegles, 1994). For the passive microwave sensors, this increased emissivity is observed through augmentation of brightness temperature (Johnson et al., 2020).

The wet-snow retrieval technique applied for this study is a statistical approach developed by Fahnestock et al. (2002) and modified by Johnson et al. (2020). The wet-snow detection is performed through a K-mean clustering algorithm. The algorithm is applied to the annual time series of brightness temperature. Wet snow is assumed when the time series shows a binomial distribution, using the criteria and thresholds defined in Johnson et al. (2020) (Figure 2).

To ensure coherency between remote sensing products and our climate model, the wet-snow masks are interpolated on the MAR grid. The grids are superimposed, and the dry/wet state for each pixel in the MAR is determined based on the most prevalent wet or dry condition observed in the corresponding area of the satellite mask. This interpolation is made with the hypothesis that the deformations and variations of the area caused by the spatial projection are negligible between a pixel and its neighbors.

### 2.1.2 Sentinel-1

One of the active sensor datasets is retrieved from the Sentinel-1 (S1) satellite constellation from the European Space Agency's (ESA) Copernicus space program. Starting with the launch of S1-A in 2014, the Sentinel-1 constellation gives access to data combining high spatial resolution and lower revisit time covering most of the globe. With the Synthetic Aperture Radar (SAR) technology, S1 products reach a spatial resolution of the order of tens of meters with a repeat pass of 6 days. By combining

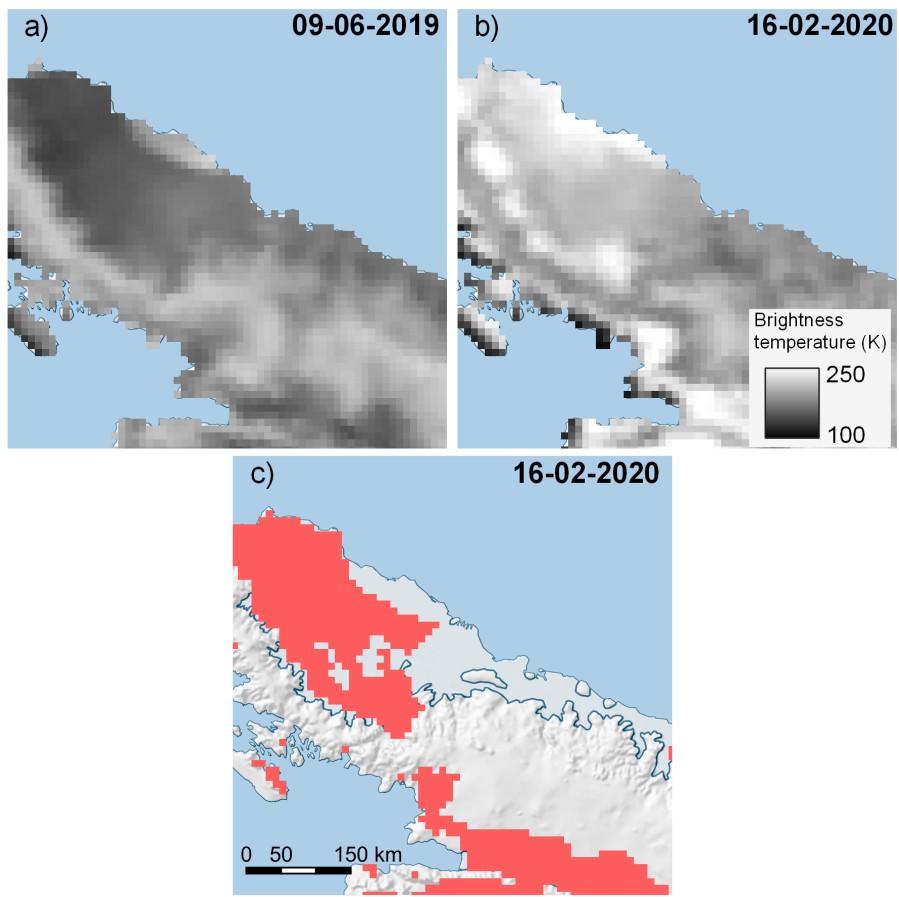

**Figure 2.** Detection of wet snow in an AMSR2 image over the Antarctic Peninsula. (a) Temperature brightness (K) the 09-06-2019. (b) Temperature brightness (K) the 16-02-2020. (c) Pixels considered as wet snow after applying the wet-snow detection algorithm. The increase in temperature brightness between (a) and (b) is attributed to the presence of liquid water in the snowpack.

different orbital paths, it is possible to reduce the time between two observations of the same location to 2-3 days in the Antarctic Peninsula. Working in C-band (5.405 GHz), it is possible to detect the presence of liquid water in the snowpack in Sentinel-1 images by identifying changes in backscattering coefficient $\sigma_0$ through time (Johnson et al., 2020). With the increase in liquid water in the snowpack, comes a change in absorptivity and scattering mechanism (Nagler and Rott, 2000). These two phenomena both lead to a decrease in the observed backscattering coefficient $\sigma_0$ (Moreira et al., 2013). As this coefficient changes little in Antarctica as long as the snowpack is dry, it is assumed that a significant change in backscattering is likely caused by the presence of water in the snowpack.

As for the passive sensors, several algorithms have been proposed to detect water in the snow with SAR and active sensors in general. Depending on the polarization, the frequency, and the nature of the snowpack, the threshold applied to the backscattering values is variable (Koskinen et al., 1997; Nagler et al., 2016). For a C-band radar, a 3-dB decrease in $\sigma_0$ has been employed

as a threshold by Nagler and Rott (2000) and Johnson et al. (2020). In the present article, we used a -2.66 dB threshold after the normalization of the images to their winter mean to classify snowpack as dry/wet. This threshold has been proposed by Liang et al. (2021) and was found to be effective on the Antarctic ice sheet.

To minimize the time between two acquisitions of Sentinel-1, all the available images were processed. To handle the quantity of data, image processing was carried out on Google Earth Engine (GEE, Gorelick et al., 2017). The S1 dataset available on GEE is already preprocessed following the implementation of the Sentinel-1 Toolbox from ESA (GEE, 2022; ESA, 2022). These processing operations include an update of the orbit metadata, removal of the low-intensity noise on the scene edges, a reduction of the discontinuities between the sub-swath, a radiometric calibration, and a terrain correction from the ASTER digital elevation model. The choice has been made to resample S1 images to a 1 km resolution using mean values before detecting wet snow as data is ultimately interpolated on the 7.5 km MAR grid. Before resampling, a 3x3 refined Lee speckle low-pass filter developed by Mullissa et al. (2021) was applied to the images in addition to a radiometric terrain flattening using the 1 arc-minute global ETOP1 DEM (Amante and Eakins, 2009). Pixels with values lower than -28 dB were removed from the dataset.

After resampling, the images are normalized to their austral winter mean. The winter mean is the average value of $\sigma_0$ for each pixel, calculated with the observations from June to October. To deal with the changes in volumetric scattering related to the acquisition geometry, only the acquisitions from the same orbit and overlapping at more than 95% are taken into account to calculate the winter mean. Consequently, differences between the acquisitions are independent of the topography and the local context. The liquid water in the snowpack is then detected in the image by applying the -2.66 dB threshold (Figure 3), following Liang et al. (2021). To create daily wet-snow masks, Sentinel-1 images of the same day were combined. In the case where three or more images overlap, the snow state is selected by a majority filter, and the acquisition time is defined as the mean time between the selected acquisitions. In the case where there are only two images that contradict each other, the non-wet status is assumed. The acquisition time selected is the acquisition time of the non-wet image.

### 2.1.3 Advanced Scatterometer

The third sensor we are using for this study is the C-band "Advanced Scatterometer" (ASCAT) aboard the MetOp satellites from the space segment of the EUMETSAT Polar System. ASCAT data are retrieved from the EUMETSAT data service portal (EUMETSAT, 2023). After resolution enhancement (Lindsley and Long, 2016), it provides a backscattering coefficient $\sigma_0$ at 4.45-km resolution by accumulating images over about 2 days. In Antarctica, only morning passes are selected for this process. The detection of the wet snow uses a simple threshold technique (Ashcraft and Long, 2006), similar to the one used for Sentinel- 1 images. The winter-mean backscattering coefficient is first calculated for each pixel and each year from the observations from June-August. Then every measurement lower than this mean -3 dB is considered wet snow. Similarly to AMSR2 daily-products, the Sentinel-1 and ASCAT daily wet/dry images are interpolated on the MAR grid. In the end, from the three satellite datasets, four binary masks have been created. One from Sentinel-1, one from ASCAT, and two from AMSR2 by splitting the ascending (evening) and the descending (morning) passes.

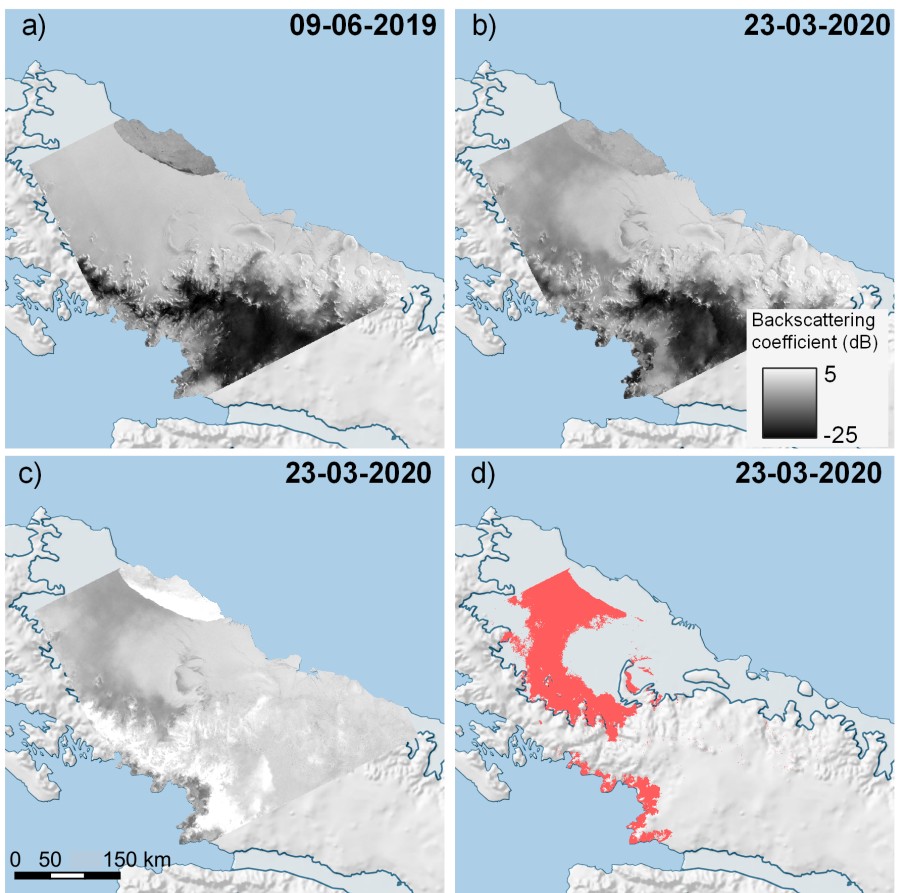

**Figure 3.** Detection of wet snow in a Sentinel-1 image over the Antarctic Peninsula. (a) Backscattering coefficient $\sigma_0$ (dB) the 09-06-2019.(b) Backscattering coefficient $\sigma_0$ (dB) the 23-03-2020. (c) Normalized backscattering coefficient of the 23-03-2020 to its winter mean. (d) Pixels considered as wet snow after thresholding the normalized image. The decrease in backscatter between (a) and (b) is attributed to the presence of liquid water in the snowpack

## 2.2   The regional climate model

We employed the Regional Climate Model MAR. MAR is a polar-oriented regional climate model mostly used to study both the Greenland (Delhasse et al., 2020; Fettweis et al., 2021) and Antarctic ice sheet (Glaude et al., 2020; Kittel et al., 2021). Its atmospheric dynamics are based on hydrostatic approximation of primitive equations originally described in Gallée and Schayes (1994) and the radiative transfer scheme is adapted from Morcrette (2002). The transfer of mass and energy between the atmospheric part of the model and the soil is handled by the Soil Ice Snow Vegetation Atmospheric Transfer module

(SISVAT, Ridder and Gallée, 1998), from which snow and ice albedo sub-modules are based on CROCUS (Brun et al., 1992). The model has been parameterized to resolve the topmost 20 meters of the snowpack, divided into 30 layers of time-varying

thickness. MAR is configured with a decreasing vertical resolution of the snow layers from the top to the bottom. The first layers are typically at the centimeter size while under the first meter, they are at the meter resolution. The four first maximum layer thicknesses are respectively 2, 5, 10, 30 cm. Each layer has a maximum water content holding capacity of 5 % of its air content beyond which the water freely percolates to the deeper layer or runoffs above impermeable layers (bare ice or ice lenses).

We employed the Regional Climate Model MAR (version 3.12). MAR is a polar-oriented regional climate model mostly used to study both the Greenland (Delhasse et al., 2020; Fettweis et al., 2021) and Antarctic ice sheets (Glaude et al., 2020; Amory et al., 2021; Kittel et al., 2021). Its atmospheric dynamics are based on hydrostatic approximation of primitive equations originally described in Gallée and Schayes (1994) and the radiative transfer scheme is adapted from Morcrette (2002). The transfer of mass and energy between the atmospheric part of the model and the soil is handled by the Soil Ice Snow Vegetation Atmospheric Transfer module (SISVAT, Ridder and Gallée, 1998), from which snow and ice albedo sub-modules are based on CROCUS (Brun et al., 1992). The model has been parameterized to resolve the top 20 first meters of the snowpack, divided into 30 layers of time-varying thickness. MAR is configured with a decreasing vertical resolution of the snow layers from the top to the bottom. The first layers are typically at the centimeter size while below the first meter, they are at the meter resolution. The four first maximum layer thicknesses are for example respectively 2, 5, 10 and respectively 30 cm. Each layer has a maximum liquid water content (LWC) of 5 % of its air content beyond which the water freely percolates to the deeper layer or runoffs above impermeable layers (bare ice or ice lenses).

For this work, the version 3.12 of MAR was used. It includes recent improvements in the snowpack temperature and the mass water conservation in the soil as described in Lambin et al. (2022). MAR was run at a 7.5 km resolution over the Antarctic Peninsula, with a 40-second time step. It was forced at its lateral boundaries and over the ocean (sea surface temperature and sea ice cover) by the 6-hourly ERA5 reanalysis (Hersbach et al., 2020) between March 2017 and May 2021. The snowpack was initialized in March 2017 with a previous MAR simulation (Kittel et al., 2021).

## 2.3 Data assimilation

The satellite sensors are sensitive to the presence of liquid water into the snowpack rather than the physical process of melt. The aim of the data assimilation is then to guide or constrain the model snowpack LWC by nudging its temperature to induce melt or refreeze to match the observed surface state (Figure 4). The assimilation routine involves comparing, pixel by pixel, the model and the satellite wet-snow masks. The satellite wet-snow mask pixel is used for the assimilation if the indicated acquisition time is separated by less than 1.5 hours from the MAR time. As up to three satellite products are assimilated at the same time, three separate cases have been developed depending on the number of assimilated masks. Each case is called according to the number of acquisitions that are taken into account in the routine. However, a daily cycle in brightness temperature and thus in wet snow can exist over Antarctica (Picard and Fily, 2006). To take it into account, if there are 3 satellite observations available for a pixel for a single day, an observation of dry snow between two wet-snow observations is considered as a false negative. Consequently, the corresponding pixel from the wet-snow masks is excluded for the day. For computational reasons,

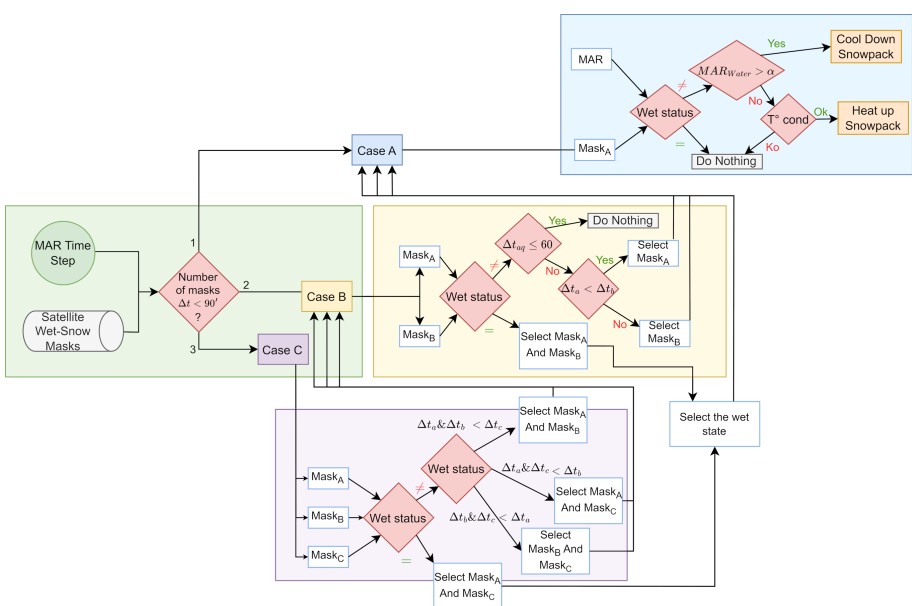

**Figure 4.** Flowchart of the assimilation algorithm. The number of satellite images available around the MAR time step determines the subprocess that is called in the routine. 3 subprocesses are defined: case A, case B, and case C. They respectively represent the availability of 1, 2, and 3 wet-snow masks for assimilation. Cases B and C are funneling to case A so no contradictory information is given to MAR.

the assimilation routine is called at each MAR time step only during the melting season, between October and April. Outside of this period, no assimilation is performed, as very little melting events are expected.

The first case of assimilation represents the situation where a single acquisition is available for a timestep (case "A" in Figure 4). It is the most frequent case applied (between 90 and 95 % of the occurrence depending on the year). This case is inspired by the assimilation performed in Kittel et al. (2022). For the 3 hours around the observation (1.5 h before the observation and 1.5 after, so the model has time to adapt its behavior but the impact remains limited), at each MAR time step, the quantity of liquid water modeled within the pixel is compared to the satellite-based mask. If the quantity of modeled LWC into the snowpack is under a certain threshold ($\alpha$) while the satellite mask indicates wet snow, the snow layers up to a certain depth ($\Delta_z$) are heated by 0.15 °C if the snow layer are colder than 0 °C. On the opposite, if LWC is above the threshold $\alpha$ but no wet snow is observed by satellites, the snowpack is cooled down by the same rate of 0.15 °C. The process is applied at each MAR time step. However, two conditions prevent change in the MAR snowpack temperature. The first is that if the snow density is above 830 $\mathrm{km\,m^{-3}}$, the layer is considered as ice and the model does not permit liquid water to accumulate into ice. The temperature is then not changed as the LWC threshold should never be reached. The second condition is the temperature of the snow layers above the $\Delta_z$. If their mean temperature is under -7.5 °C, the MAR snowpack is too cold to be able to produce meltwater in the model by warming its snowpack, and the satellite observation is ignored. This operation is repeated until the $\alpha$ threshold is reached or the observation is out of the time range. The choices for thresholds $\alpha$ and $\Delta_z$ are discussed in the two next sections, 2.3.1 and 2.3.2.

The second case is called when there are two satellite observations at less than 1.5 hours from MAR time (case "B" in Figure 4). If the two masks agree, the two observations are associated with the first case but with a $\Delta_z$ equivalent to the mean values of the thresholds that would have been used for individual masks. If the two observations indicate different snow states, a different processing is applied if the acquisitions are close to each other in time (within an hour) or not. For two inconsistent observations spread by more than one hour, the assimilated snow state is the snow state from the closest image to the MAR time, following the first case. For two close contradictory observations, nothing is assimilated as they are considered both equally likely to be correct or wrong. Valuable information may be lost in this case. The difference in penetration depth can cause a deeper penetrating signal to observe liquid water (Figure 5). However, as we have no additional information on the depth at which the water may be present, the model is run as if there was no observation available.

The second case occurs when there are only two satellite observations at less than 1.5 hours from MAR time (case "B" in Figure 4). If the two masks agree, the two observations are associated with the first case but with a depth $\Delta_z$ equivalent to the mean values of the thresholds that would have been used for individual masks. If the two observations indicate different snow states, a different processing is applied if the acquisitions are close to each other in time (within an hour) or not. For two inconsistent observations spread by more than one hour, the assimilated snow state is the snow state from the closest image to the MAR time, following the first case (Case "A"). For two close contradictory observations, nothing is assimilated as they are considered both equally likely to be correct or wrong. Valuable information may be lost in this case. The difference in penetration depth can cause a deeper penetrating signal to observe liquid water (Figure 5). However, as we have no additional information on the depth at which the water may be present, the model is run as if there was no observation available.

The third case is when all three observations are available within the same 3-hour time window (case "C" in Figure 4). As for the second case, if the three masks agree with the same wet/non-wet snow status, they are considered as one and the first case is called. Again, the depth $\Delta_z$ used is equivalent to the mean values of the thresholds that would have been used separately. If an observation is different from the other two, the two closest observations of the MAR time are analyzed using the second case described here above. For our configuration of sensors, this third case is only encountered a couple of times (less than 1 % of all occurrences) while assimilating wet-snow masks of AMSR2 (ascending orbit), ASCAT, and Sentinel-1.

### 2.3.1 Choice of water content threshold ($\alpha$)

Estimating the quantity of liquid water into the snowpack with a single satellite acquisition is challenging. Despite the numerous research studies, the knowledge on the subject remains limited (Trusel et al., 2013; Fricker et al., 2021). However, as described in Picard et al. (2022), it is possible to find a typical water quantity from which the satellite signal significantly changes and can be detected as melting/wet snow. Picard et al. (2022) demonstrates the capability of detecting little amounts of water using the radio frequencies employed in this study. Only 0.11 and 0.05 $\mathrm{kg\,m^{-2}}$ of liquid water is necessary at respectively 6 GHz and 19 GHz if the water is uniformly spread over the pixel. This quantity can be higher for heterogeneous pixels containing dry/wet patches. For this study, the choice has been made to use the same threshold no matter the sensor frequency. AMSR2 acquires data at higher frequencies and is theoretically more sensitive, but it has a coarser resolution than the two active sensors. Its pixels tend to be more heterogeneous, suggesting a compensation in liquid water quantity. Two different thresholds are tested

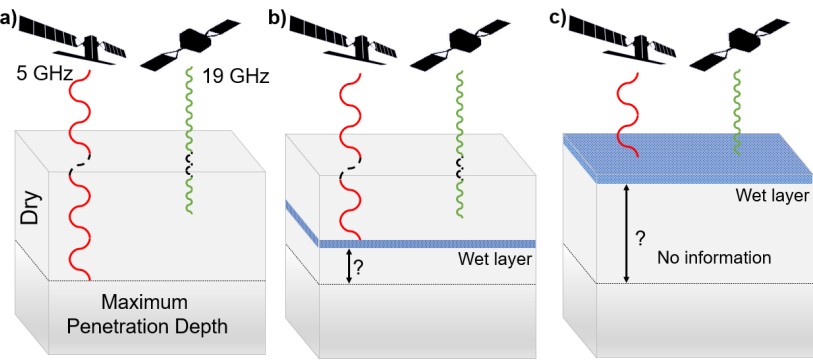

**Figure 5.** Illustration of the penetration depth of the microwave sensor according to their wavelength and the depth of the wet snow layer. (a) Penetration depth in a dry snowpack. The signal of the sensor with the lower frequency (5 GHz, in red) penetrates deeper than the signal of the higher one (19 GHz, in green). (b) Penetration depth with a layer of liquid water deep in the snowpack. The microwave sensor with deeper penetration can detect water presence but the other cannot. (c) Penetration depth with liquid water at the top of the snowpack. Both satellites can observe the presence of liquid water.

to study the sensitivity of the model. Both have been shown to significantly change the snowpack brightness temperature in the literature. Tedesco et al. (2007) proposed a LWC threshold of 0.2 % while Picard et al. (2022) proposed 0.1 % of the snowpack mass being liquid water. They both have already been tested in Kittel et al. (2022) where the choice between the two was found not to significantly influence the melt quantity produced by the MAR model. The sensitivity of the microwave is high enough that the quantities of liquid water that can be detected are much smaller than that produced during a typical melting day ($1.2 \pm 0.6\%$ as modeled by MAR over the studied zone in the top meter of snow). Currently, there is no clue to identify the best-fitting threshold for this study.

### 2.3.2 Choice of assimilation depth threshold ($\Delta_z$)

Microwaves have penetration capabilities directly related to their wavelength (Elachi and van Zyl, 2006). As a consequence, C-band from Sentinel-1 and ASCAT has a different penetration depth than Ku-band from AMSR2. In addition, the water content strongly influences the penetration depth, as water at the top of the snowpack can prevent deeper penetration (Figure 5). In this experiment, we set different penetration depths for each remote-sensing product to test its influence. Using AMSR2 (Ku-Band), we consider a depth $\Delta_z = 0.1, 0.2$, and 0.4 m successively below the surface. Below this depth, the electromagnetic wave should not have a noticeable influence (Picard et al., 2022). For Sentinel-1 and ASCAT (C-band), the depth thresholds $\Delta_z$ are set up to 0.5, 1, and 1.5 m, as the signal is expected to penetrate deeper in the snowpack.

### 2.3.3 Experiences conducted

An ensemble of 24 MAR simulations is presented here. Only the reference MAR simulation, $MAR_{ref}$, is performed without assimilation. The others are referred to as "assimilations" hereafter. For each one, the satellite wet-snow masks are assimilated into the model, with different parameters (Table 2). The reference assimilation ($Assim_{ref}$) is using Sentinel-1 and AMSR2, both their ascending and descending orbits, and with thresholds $\Delta_z$ = 1 m and 0.2 m respectively. Also, $\alpha$ is set at 0.1 %. The thresholds used to perform $Assim_{ref}$ correspond to values given in the literature (Elachi and van Zyl, 2006; Picard et al., 2022). The other assimilations have been performed with a combination of 3 satellite products chosen between Sentinel-1, AMSR2 ascending, AMSR2 descending, and ASCAT, and with a combination of the assimilation parameters. The assimilations have been performed from June 2019 to May 2020, and from June 2020 to May 2021.

An ensemble of 24 MAR simulations is presented here (Table 2). Only the reference MAR simulation, $MAR_{ref}$, is performed without assimilation. The others are referred to as "assimilations" hereafter. Their naming convention is "ASA" followed by the value of $\alpha$ threshold in subscript (in %) and the RS datasets assimilated and their corresponding $\Delta_z$ threshold value in subscript (in m). "S1" refers to the S1 dataset, "AMA" to AMSR2 ascending, "AMD" to AMSR2 descending, and "AS" to ASCAT. The assimilations were started in January 2019, and have been restarted from the simulation without assimilation (initialized in 2017). For each one, the satellite wet-snow masks are assimilated into the model, with different parameters (Table 2). The reference assimilation ($Assim_{ref}$) is using Sentinel-1 and AMSR2, both their ascending and descending orbits, and with assimilation depth thresholds $\Delta_z$ = 1 m and 0.2 m respectively. Also, the liquid water content threshold $\alpha$ is set at 0.1 %. The thresholds used to perform $Assim_{ref}$ correspond to values given in the literature (Elachi and van Zyl, 2006; Picard et al., 2022). The other assimilated simulations have been performed with a combination of 3 satellite products chosen between Sentinel-1, AMSR2 ascending, AMSR2 descending, and ASCAT, and with a combination of the assimilation parameters. The present document focuses on the 2019-2020 melt season, while the 2020-2021 season graphs and tables are available in Supplementary Materials.

## 3 Evaluation

Because the integrated physics within RCMs is either partially resolved or contains uncertainties, it is first required to evaluate model outputs to in situ measurements. The evaluation is there to quantify how close the model is to reality and if the model is inclined to reproduce this observed situation. Since our focus is on assessing the model sensitivity through assimilation, we exclusively evaluate MAR without assimilation. It is worth noting that the values derived from assimilations may diverge from the observations due to the assimilation algorithm sensitivity rather than the model physics.

The outputs of the non-assimilated model are evaluated by comparing with in situ observations. The daily observations are provided by Automatic Weather Stations (AWS) widespread across the AIS. Here, 9 weather-stations datasets available in the studied zone (blue crosses displayed in Figure 1) have been gathered to calculate statistics between the model and the observations as done in Kittel (2021) and Mottram et al. (2021). The statistics employed for the evaluation are the Mean Bias (MB), Root Mean Square Error (RMSE), Centered Root Mean Square Error (CRMSE), and correlation (r) (Table 3).

**Table 2.** Name of the different simulations and parameterization of the simulation with data assimilation. When not mentioned, both ascending and descending paths of AMSR2 are assimilated. Simulations marked with an asterisk and one sensor assimilation are not taken into account in the calculation of the ensemble average.

| Name | $\alpha$ (%) | Ku-band $\Delta_z$ (m) | C-band $\Delta_z$ (m) | Sensors |
|---|---|---|---|---|
| $Assim_{ref}$ | 0.1 | 0.2 | 1 | AMSR2 + S1 |
| $AsA_{01}S1_{05}AMA_{02}AMD_{02}$ | 0.1 | 0.2 | 0.5 | AMSR2 + S1 |
| $AsA_{01}S1_{15}AMA_{02}AMD_{02}$ | 0.1 | 0.2 | 1.5 | AMSR2 + S1 |
| $AsA_{02}S1_{10}AMA_{02}AMD_{02}$ | 0.2 | 0.2 | 1 | AMSR2 + S1 |
| $AsA_{02}S1_{05}AMA_{02}AMD_{02}$ | 0.2 | 0.2 | 0.5 | AMSR2 + S1 |
| $AsA_{02}S1_{15}AMA_{02}AMD_{02}$ | 0.2 | 0.2 | 1.5 | AMSR2 + S1 |
| $AsA_{01}S1_{10}AMA_{01}AMD_{01}$ | 0.1 | 0.1 | 1 | AMSR2 + S1 |
| $AsA_{01}S1_{05}AMA_{01}AMD_{01}$ | 0.1 | 0.1 | 0.5 | AMSR2 + S1 |
| $AsA_{01}S1_{15}AMA_{01}AMD_{01}$ | 0.1 | 0.1 | 1.5 | AMSR2 + S1 |
| $AsA_{02}S1_{10}AMA_{01}AMD_{01}*$ | 0.2 | 0.1 | 1 | AMSR2 + S1 |
| $AsA_{02}S1_{05}AMA_{01}AMD_{01}*$ | 0.2 | 0.1 | 0.5 | AMSR2 + S1 |
| $AsA_{02}S1_{15}AMA_{01}AMD_{01}*$ | 0.2 | 0.1 | 1.5 | AMSR2 + S1 |
| $AsA_{01}S1_{10}AMA_{04}AMD_{04}$ | 0.1 | 0.4 | 1 | AMSR2 + S1 |
| $AsA_{01}S1_{05}AMA_{04}AMD_{04}$ | 0.1 | 0.4 | 0.5 | AMSR2 + S1 |
| $AsA_{01}S1_{15}AMA_{04}AMD_{04}$ | 0.1 | 0.4 | 1.5 | AMSR2 + S1 |
| $AsA_{02}S1_{10}AMA_{04}AMD_{04}$ | 0.2 | 0.4 | 1 | AMSR2 + S1 |
| $AsA_{02}S1_{05}AMA_{04}AMD_{04}$ | 0.2 | 0.4 | 0.5 | AMSR2 + S1 |
| $AsA_{02}S1_{15}AMA_{04}AMD_{04}$ | 0.2 | 0.4 | 1.5 | AMSR2 + S1 |
| $AsA_{01}S1_{10}AMA_{02}AS_{02}$ | 0.1 | 0.2 | 1 | AMSR2 (asc.) + S1 + ASCAT |
| $AsA_{01}AMA_{02}$ | 0.1 | 0.2 | / | AMSR2 (asc.) |
| $AsA_{01}AMD_{02}$ | 0.1 | 0.2 | / | AMSR2 (desc.) |
| $AsA_{01}S1_{10}$ | 0.1 | / | 1 | S1 |
| $AsA_{01}AS_{10}$ | 0.1 | / | 1 | ASCAT |
| $MAR_{ref}$ | / | / | / | None |

The statistics are listed for the 2016-2021 period for the near-surface pressure, temperature, wind speed, relative humidity, and modeled energy-balance components, including short-wavelength downward radiations (SWD), short-wavelength upward radiations (SWU), long- wavelength downward radiations (LWD), and long-wavelength upward radiations (LWU).

Small biases can exist between the in-situ observations and the model due to the elevation difference. The AWS observations are punctual when the model provides zonal information over a 7.5 x 7.5 $km^2$ pixel. Thus the mean elevation of the MAR pixel in which the AWS falls is not the same as the AWS true elevation. This difference is particularly noticeable for the near-surface

**Table 3.** Mean Bias (MB), Root Mean Square Error (RMSE), Centered Root Mean Square Error (CRMSE), and correlation between MAR and daily observation over the Antarctic Peninsula. A negative value implies a lower MAR estimate than the observation. Statistics are given for the near-surface pressure, temperature, wind speed, relative humidity, shortwave downward (SWD), shortwave upward (SWU), longwave downward (LWD), and longwave upward (LWU) annually, for the summer (DJF), and for the winter (JJA) and are calculated for the 2016-2021 period. During winter, the absence of the Sun implies no short-wavelength solar radiation measurements (SWD and SWU). Locations of the weather station used for the daily observations are marked by blue crosses in Figure 1.

| | Annual | | | | Summer | | | | Winter | | | |
|---|---|---|---|---|---|---|---|---|---|---|---|---|
| | MB | RMSE | CRMSE | Correlation | MB | RMSE | CRMSE | Correlation | MB | RMSE | CRMSE | Correlation |
| Near Surface Pressure (hPa) | -5.44 | 14.57 | 1.25 | 0.99 | -5.69 | 13.18 | 0.87 | 0.99 | -6.13 | 16.09 | 1.42 | 0.99 |
| Temperature (°C) | -0.32 | 3.32 | 2.81 | 0.93 | -1.13 | 2.36 | 1.68 | 0.76 | 0.3 | 3.63 | 3.11 | 0.92 |
| Wind speed ($\mathrm{m\,s^{-1}}$) | -0.39 | 2.58 | 2.28 | 0.79 | -0.43 | 2.22 | 1.85 | 0.7 | -0.35 | 2.92 | 2.57 | 0.78 |
| Relative humidity (%) | 3.2 | 8.73 | 8.13 | 0.72 | 6.88 | 9.32 | 6.29 | 0.75 | 2.87 | 9.1 | 8.64 | 0.79 |
| SWD ($\mathrm{W\,m^{-2}}$) | 13.87 | 36.23 | 33.46 | 0.97 | 41.58 | 59.21 | 42.15 | 0.79 | / | / | / | / |
| SWU ($\mathrm{W\,m^{-2}}$) | -0.2 | 24.04 | 24.04 | 0.97 | 14.38 | 35.81 | 32.8 | 0.78 | / | / | / | / |
| LWD ($\mathrm{W\,m^{-2}}$) | -14.75 | 26.15 | 21.59 | 0.76 | -26.56 | 32.51 | 18.75 | 0.65 | -7.12 | 21.08 | 19.85 | 0.81 |
| LWU ($\mathrm{W\,m^{-2}}$) | 3.4 | 14.2 | 13.79 | 0.93 | -0.52 | 9.2 | 9.19 | 0.76 | 2.83 | 17.12 | 16.88 | 0.9 |

pressure, directly linked to the elevation. Nonetheless, a high correlation (r > 0.98) reflects the ability to simulate its temporal variability.

In general, the winter season is slightly better represented with higher correlations and lower mean bias than the summer season. A weaker correlation is observed in summer for long-wavelength downward radiations (r = 0.65). This difference is compensated by the excess of short-wavelength solar radiations in summer. MAR does not assimilate temperature profile nor coastal temperature but is only forced at its lateral boundaries every 6h for its specific humidity and temperature. Thus modeled clouds are the outcome of the model climate and microphysics Delhasse et al. (2020). Moreover, the radiative scheme implemented in MAR is the one from the ERA-40 reanalysis. This scheme has been updated in the ERA-5 reanalysis (Hersbach et al., 2020) but not in the model. MAR underestimates the liquid water path during summer when compared to Cloudsat-CALISPO estimates described in (Van Tricht et al., 2016). Such underestimation is partially responsible for the LWD bias in summer.

In addition, Jakobs et al. (2020) provide melt estimates from the AWS. These estimates can be compared to the surface melt production of the four closest MAR pixels of the AWS (Figure 6). MAR tends to overestimate some extremes of melting while simultaneously underestimating or overestimating the duration of periods during which the ice shelves are experiencing melting. Even though there can be a difference in altitude between the AWS and MAR pixels that explains the differences between the two datasets, these discrepancies also highlight the importance of nudging MAR to correspond to the remote sensing observation of wet snowpack.

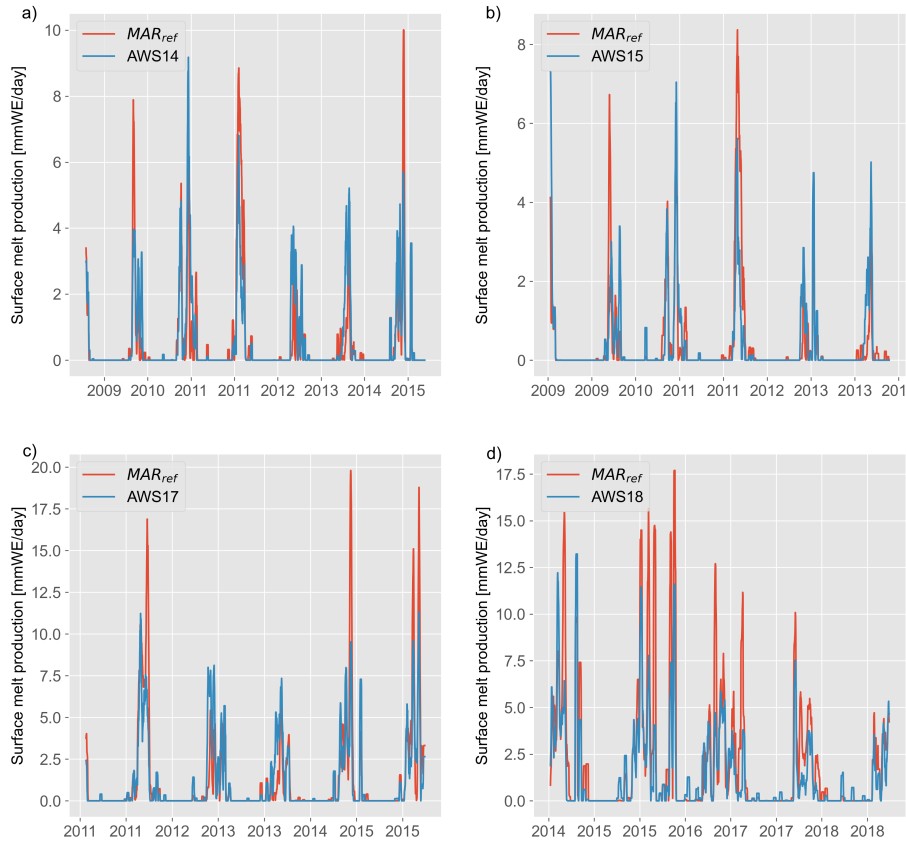

**Figure 6.** Comparison of surface melt production ($\mathrm{mmWEday}^{-1}$) as modeled by $MAR_{ref}$ (in red) and estimated surface melt production from AWS (a) 14, (b) 15, (c) 17, and (d) 18 (in blue) described in Jakobs et al. (2020). In comparison to the AWS, MAR tends to overestimate peaks of melt but underestimate smaller melt seasons.

## 4 Results

Table 4 provides a comprehensive summary of the results obtained from the 24 MAR simulations. The summary includes the number of melt days (*i.e.* the number of days where melt is occurring over at least 10 % of the studied zone), surface Melt (ME), Runoff (RU), Refreeze (RZ), and the Surface Mass Balance (SMB). This table offers a concise overview of the simulation results. In the case of assessing the sensitivity of the MAR model to the assimilation, we analyzed the evolution of several variables (Table 5) including ME, RU, SMB, Snowpack Density ($\rho$), and Liquid Water Content (LWC) to study the impact caused by the data assimilation. The first 4 variables (ME, RU, RZ, and SMB) are given for the entire snowpack profile while the other two $\rho$ and LWC are given for the first meter. The average value of the variables of all the assimilations, $\overline{Assim}$, is compared to the model with no assimilation. Although $\overline{Assim}$ differs from the reference assimilation, $Assim_{ref}$ is the closest simulation to $\overline{Assim}$. Three simulations have been discarded to calculate $\overline{Assim}$ because of the unrealistic freeze/thaw

**Table 4.** Summary of the results of the different experiments conducted for the study. The number of melt days, cumulated surface meltwater, runoff, refreeze, and surface mass balance over the 2019-2020 melt season are provided for each experiment.

| Simulation | Number of melt days | ME (Gt yr$^{-1}$) | RU (Gt yr$^{-1}$) | RZ (Gt yr$^{-1}$) | SMB (Gt yr$^{-1}$) |
|---|---|---|---|---|---|
| $Assim_{ref}$ | 121 | 214 | 56 | 182 | 427 |
| $AsA_{01}S1_{05}AMA_{02}AMD_{02}$ | 123 | 214 | 55 | 181 | 429 |
| $AsA_{01}S1_{15}AMA_{02}AMD_{02}$ | 121 | 213 | 55 | 180 | 429 |
| $AsA_{02}S1_{10}AMA_{02}AMD_{02}$ | 129 | 297 | 59 | 256 | 425 |
| $AsA_{02}S1_{05}AMA_{02}AMD_{02}$ | 129 | 299 | 58 | 258 | 426 |
| $AsA_{02}S1_{15}AMA_{02}AMD_{02}$ | 126 | 298 | 60 | 256 | 424 |
| $AsA_{01}S1_{10}AMA_{01}AMD_{01}$ | 122 | 293 | 56 | 257 | 428 |
| $AsA_{01}S1_{05}AMA_{01}AMD_{01}$ | 123 | 289 | 48 | 258 | 436 |
| $AsA_{01}S1_{15}AMA_{01}AMD_{01}$ | 121 | 288 | 51 | 255 | 432 |
| $AsA_{02}S1_{10}AMA_{01}AMD_{01}$ | 130 | 604 | 186 | 430 | 298 |
| $AsA_{02}S1_{05}AMA_{01}AMD_{01}$ | 131 | 626 | 203 | 433 | 280 |
| $AsA_{02}S1_{15}AMA_{01}AMD_{01}$ | 126 | 581 | 177 | 418 | 307 |
| $AsA_{01}S1_{10}AMA_{04}AMD_{04}$ | 120 | 184 | 45 | 161 | 438 |
| $AsA_{01}S1_{05}AMA_{04}AMD_{04}$ | 123 | 186 | 47 | 163 | 437 |
| $AsA_{01}S1_{15}AMA_{04}AMD_{04}$ | 120 | 183 | 45 | 161 | 439 |
| $AsA_{02}S1_{10}AMA_{04}AMD_{04}$ | 127 | 214 | 53 | 184 | 431 |
| $AsA_{02}S1_{05}AMA_{04}AMD_{04}$ | 129 | 221 | 56 | 187 | 427 |
| $AsA_{02}S1_{15}AMA_{04}AMD_{04}$ | 126 | 213 | 52 | 183 | 431 |
| $AsA_{01}S1_{10}AMA_{02}AS_{02}$ | 122 | 191 | 47 | 167 | 436 |
| $AsA_{01}AMA_{02}$ | 121 | 177 | 48 | 153 | 436 |
| $AsA_{01}AMD_{02}$ | 120 | 143 | 39 | 128 | 445 |
| $AsA_{01}S1_{10}$ | 119 | 148 | 39 | 131 | 444 |
| $AsA_{01}AS_{10}$ | 121 | 155 | 41 | 137 | 442 |
| $MAR_{ref}$ | 123 | 142 | 32 | 132 | 451 |

cycle induced by the assimilation. These simulations are marked with an asterisk in Table 2. Not to include bias from one wet-snow mask, simulations assimilating only one wet-snow mask are also not used in the calculation of $\overline{Assim}$.

The surface melt production is larger for all assimilations, compared to $MAR_{ref}$. On average, the wet-snow extent provided by the wet-snow masks is larger than the extent modeled by $MAR_{ref}$ on the Antarctic Peninsula. This difference impacts the melt production in the model. No matter the parametrization of the assimilation, the surface melt production is increased compared to $MAR_{ref}$ (Table 5), leading to a cumulated melt production increase of 66.7 % for $\overline{Assim}$ over the year.

The meltwater will eventually either refreeze or runoff, depending on the saturation level of the snowpack. The snowpack can saturate, either from excess in meltwater production or from densification. If snowpack LWC exceeds 5 % of the firn air

**Table 5.** Difference (in %) in surface melt production (ME), runoff (RU), refreeze (RZ), surface mass balance (SMB), snowpack liquid water content (LWC), and snowpack density ($\rho$) between $MAR_{ref}$ and the mean value of the assimilations ($\overline{Assim}$) over the Antarctic Peninsula in 2019 - 2020. Variables are cumulated annually and over summer (from November to the end of April) except for snowpack density and the liquid water content which are averaged over the periods. LWC and $\rho$ are given as for the average of the snowpack first meter while the other variables are cumulated on the whole modeled snowpack.

| | Annual | | | | | Summer | | | | |
|---|---|---|---|---|---|---|---|---|---|---|
| | $MAR_{ref}$ | $Assim_{ref}$ | $\overline{Assim}$ | Range | Difference (%) | $MAR_{ref}$ | $Assim_{ref}$ | $\overline{Assim}$ | Range | Difference (%) |
| ME $(\mathrm{Gt\,yr^{-1}})$ | 142 | 214 | 237 | 183 - 299 | 66.7 | 140 | 212 | 235 | 180 - 296 | 67.1 |
| RU $(\mathrm{Gt\,yr^{-1}})$ | 32 | 56 | 53 | 45 - 60 | 63.8 | 32 | 56 | 53 | 45 - 60 | 64.5 |
| RZ $(\mathrm{Gt\,yr^{-1}})$ | 132 | 182 | 206 | 161 - 258 | 55.7 | 128 | 176 | 201 | 157 - 253 | 56.5 |
| SMB $(\mathrm{Gt\,yr^{-1}})$ | 451 | 427 | 431 | 424 - 439 | -4.5 | 253 | 229 | 233 | 226 - 240 | -8.2 |
| $LWC_{1\mathrm{m}}$ $(\mathrm{g\,kg^{-1}})$ | 19 | 17 | 18 | 14 - 24 | -6.4 | 33 | 29 | 31 | 24 - 40 | -6 |
| $\rho_{1\mathrm{m}}$ $(\mathrm{kg\,m^{-3}})$ | 407 | 422 | 421 | 418 - 424 | 3.6 | 425 | 445 | 445 | 440 - 449 | 4.6 |

content, the excess water starts to trickle and runoff (irreducible water saturation). The evolution of runoff is thus directly related to the evolution of melt and the snowpack saturation level (Figure 7). Therefore, the relative increase in surface melt and runoff is almost similar between $\overline{Assim}$ and $MAR_{ref}$ (66.7% and 63.8%, respectively) but, their absolute increase is not the same (+95 $\mathrm{Gt\,y^{-1}}$ and +21 $\mathrm{Gt\,y^{-1}}$, respectively).

The difference between the increase in meltwater production and the increase in runoff corresponds to the increase in refreezing. This suggests that the snowpack can still absorb liquid water unless it reaches its maximum LWC. The strongest increase in runoff occurs together with firn air content depletion over the ice shelves. Liquid can stay in the porous layers of the surface snowpack. Then, depending on the available energy in the system, the water either refreezes during the following night or percolates deeper in the snowpack. But, by refreezing, the water densifies the firn, causing firn air content depletion, leaving less storage space for liquid water in the perennial snowpack (Banwell et al., 2021).

As it can be seen in Figure 8, the data assimilation only has a slight effect on the SMB. The SMB expression is defined as the sum of the ablation terms (runoff and sublimation) and accumulation terms (snowfall and rainfall). The cumulated SMB for the 2019-2020 melt season is only decreased by 4.5 % compared to the model without assimilation. The general trend of SMB remains positive in the studied zone. Only the ice shelves show negative SMB during austral summer (Figure 9).

The density and LWC of the snowpack are also impacted by the assimilation. As presented in Table 6, on the ice shelves, where most of the surface melt and refreezing occurs, densification affects the LWC. With a denser snowpack, firn air content is reduced and there is less space for liquid water to be absorbed. Therefore, despite the increase in surface melt production, the assimilation process eventually led to a decrease in the amount of liquid water retained in the snowpack. This reduction occurs due to the assimilations impact on water retention capabilities of the snowpack.

All three highlighted ice shelves (Larsen C, Wilkins, and George VI) are experiencing an increase in surface melt, refreeze, and runoff (Table 6). On Larsen C and Wilkins ice shelves, the increase in runoff is strongly superior to the increase in surface melt production. Larsen C is the ice shelf experiencing the higher increase of melt in absolute and relative (+21 $\mathrm{Gt\,y^{-1}}$, *i.e.*

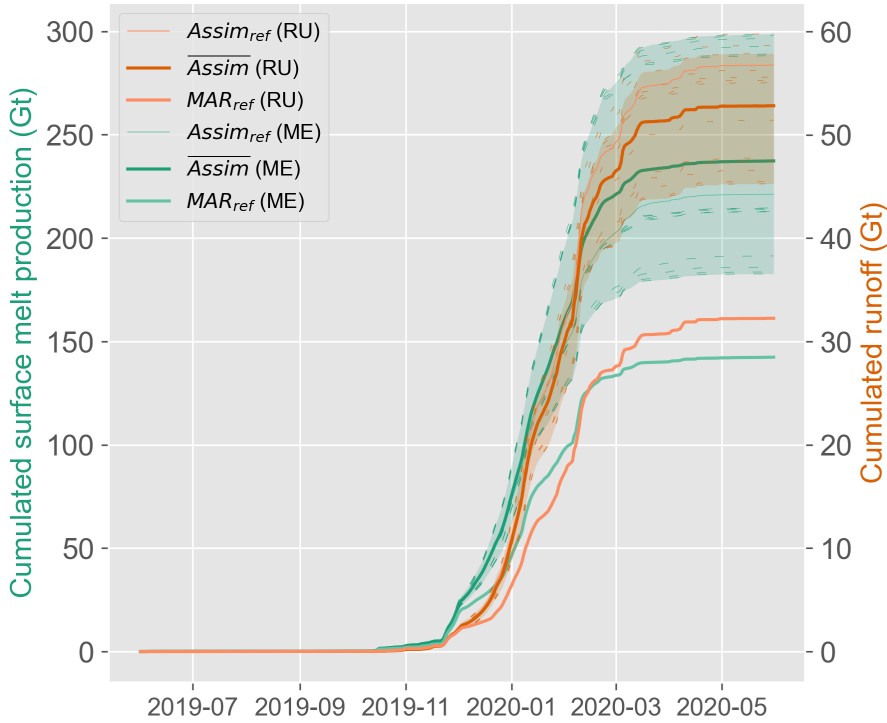

**Figure 7.** Comparison between the cumulated surface melt (Gt) in green and the cumulated runoff (Gt) in orange for the 2019- 2020 melt season as modeled by MAR without assimilation and with data assimilation. Shaded areas represent the range of the assimilations. While the increase in Gt is larger for melt production, the relative increase is mostly the same for melt production and runoff.

+85.7 %) of the three, and its runoff is tripled (+6 $\mathrm{Gt\,y^{-1}}$, *i.e.* +311.2 %). However, over the year, its liquid water content tends to slightly increase (+1%). It would therefore seem that on ice shelves, the increase in refreezing is not strong enough to compensate for the increase in melting. The depletion of firn air content leads to a swift saturation of the snowpack, making

the surplus of meltwater resulting in a more pronounced decrease in SMB compared to other regions of the AP.

All three highlighted ice shelves (Larsen C, Wilkins, and George VI) are experiencing an increase in surface melt, refreeze, and runoff (Table 6). On Larsen C and Wilkins ice shelves, the increase in runoff is strongly superior to the increase in surface melt production. Larsen C is the ice shelf experiencing the higher increase of melt in absolute and relative (+21 $\mathrm{Gt\,y^{-1}}$, i.e. +85.7 %) of the three, and its runoff is tripled (+6 $\mathrm{Gt\,y^{-1}}$, i.e. +311.2 %). However, over the year, its liquid water content

tends to slightly increase (+1 %). It would therefore seem that on ice shelves, the increase in refreezing is not strong enough to compensate for the increase in melting. The depletion of firn air content leads to a swift saturation of the snowpack, making the surplus of meltwater resulting in a more pronounced decrease in SMB compared to other regions of the Antarctic Peninsula.

Except for the LWC, which remains relatively small and stable as it has been averaged over the season, the analyzed variables (ME, RU, RZ, SMB, and snowpack density) have undergone noticeable variations, causing $MAR_{ref}$ variables to always be

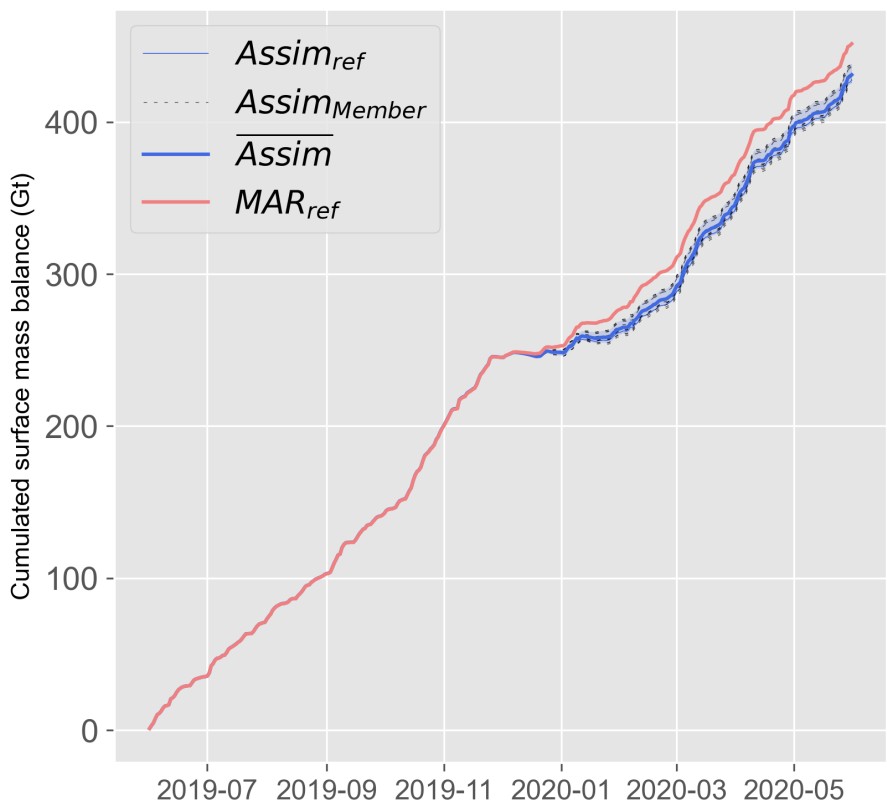

**Figure 8.** Cumulated surface mass balance (Gt) for 2019-2020 melt season as modeled by MAR without assimilation ($MAR_{ref}$ in red), with data assimilation ($Assim_{member}$ in dashed lines), and their averaged value ($\overline{Assim}$ in blue). Shaded areas represent the range of the assimilations. Despite the increase in surface melt, the surface mass balance does not significantly decrease.

outside of the assimilated-simulations range during summer. The amplified surface melt production leads to concurrent effects, including increased runoff, reduced surface mass balance, and an increased occurrence of refreezing. This increase in runoff is attributed to the compaction of the upper layers of the snowpack, which reduces its capacity to absorb meltwater.

     In the end, the results illustrate that, on average, $Assim_{ref}$ is the assimilation that gives the closest results to $\overline{Assim}$ and makes it an appropriate candidate when computational resources are limited (one simulation instead of 24). If the sensitivity to 385    the different parameters of the assimilation is discussed hereafter, the parameters used in $Assim_{ref}$ seem to be an appropriate option.

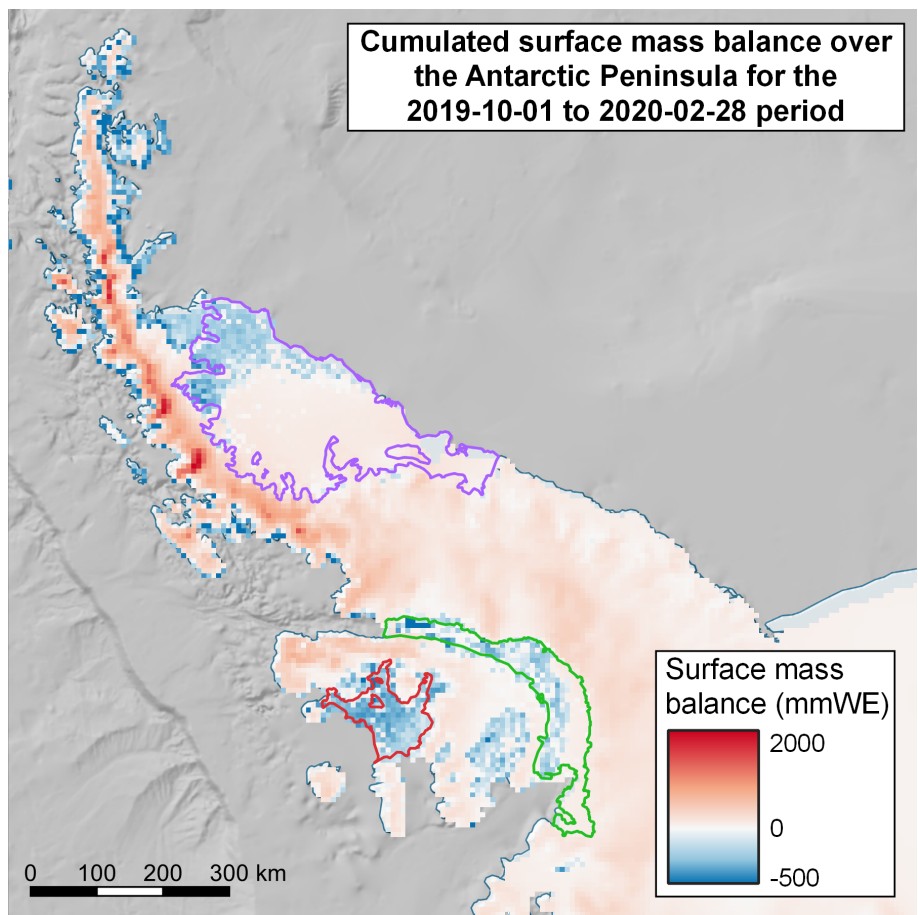

**Figure 9.** Cumulated SMB (mmWE) from 2019-10-01 to 2020-02-28 over the AP as modeled by $Assim_{ref}$. Larsen C is outlined in purple, George VI in green, and Wilkins in red. The southern ice shelves and the northernmost coastlines are experiencing a negative SMB in opposition to the rest of the AP. Larsen C is divided in two. Its northern part is experiencing a negative SMB while the southern part is positive.

## 4.1 MAR sensitivity

### 4.1.1 Assimilation depth threshold sensitivity

The assimilation depth of low penetrating sensors influences melt production by inducing firn air content depletion. Due to refreezing, the uppermost 10 centimeters of the snowpack becomes denser compared to the top meter. The refereeze is accentuated when using a shallow-depth threshold (for example 10 centimeters with AMSR2) as the top layers of the snowpack will contain the majority of the liquid water. Consequently, the increase in melt production to reach the $\alpha$ threshold (0.1% or 0.2%) will be greater than for a deeper assimilation depth where less densification occurs. Also, with firn air content depletion, two other phenomena enhance melt production. First, the available energy in the system is consumed by the melting process,

**Table 6.** Difference (%) in surface melt (ME), runoff (RU), refreeze (RZ), surface mass balance (SMB), snowpack liquid water content (LWC), and snowpack density ($\rho$) between $MAR_{ref}$ and the mean value of the assimilations ($\overline{Assim}$) over the three highlighted ice shelves in 2019 - 2020. Variables are cumulated annually and over summer (from November to the end of April) except for snowpack density and the liquid water content which are averaged over the periods. LWC and $\rho$ are given as for the average of the snowpack first meter while the other variables are cumulated on the whole modeled snowpack.

| Larsen C | Annual | | | | | Summer | | | | |
|---|---|---|---|---|---|---|---|---|---|---|
| | $MAR_{ref}$ | $Assim_{ref}$ | $\overline{Assim}$ | Range | Difference (%) | $MAR_{ref}$ | $Assim_{ref}$ | $\overline{Assim}$ | Range | Difference (%) |
| ME (Gt yr$^{-1}$) | 23 | 38 | 44 | 31 - 58 | 85.7 | 23 | 38 | 43 | 30 - 57 | 87.6 |
| RU (Gt yr$^{-1}$) | 2 | 7 | 8 | 4 - 10 | 311.2 | 2 | 7 | 8 | 4 - 10 | 311.6 |
| RZ (Gt yr$^{-1}$) | 22 | 32 | 36 | 28 - 50 | 62.2 | 22 | 31 | 36 | 27 - 49 | 63.6 |
| SMB (Gt yr$^{-1}$) | 24 | 19 | 18 | 15 - 21 | -25.1 | 15 | 10 | 9 | 6 - 13 | -38.9 |
| $LWC_{1m}$ (g kg$^{-1}$) | 3.6 | 3.5 | 3.6 | 3.1 - 4.6 | 1.5 | 6.1 | 6.0 | 6.2 | 5.2 - 7.8 | 1.1 |
| $\rho_{1m}$ (kg m$^{-3}$) | 463 | 508 | 509 | 495 - 519 | 9.8 | 500 | 549 | 552 | 536 - 564 | 10.3 |
| **Wilkins** | | | | | | | | | | |
| ME (Gt yr$^{-1}$) | 9 | 13 | 14 | 10 - 19 | 48.4 | 9 | 12 | 14 | 10 - 19 | 48.2 |
| RU (Gt yr$^{-1}$) | 2 | 5 | 4 | 2 - 7 | 185.6 | 2 | 5 | 4 | 2 - 7 | 185.6 |
| RZ (Gt yr$^{-1}$) | 9 | 9 | 11 | 9 - 15 | 22.2 | 9 | 8 | 10 | 8 - 15 | 21 |
| SMB (Gt yr$^{-1}$) | 6 | 2 | 3 | 0 - 5 | -51.3 | 2 | -2 | -1 | -4 - 1 | -141.4 |
| $LWC_{1m}$ (g kg$^{-1}$) | 1.3 | 1.0 | 1.0 | 1.0 - 1.21 | -21 | 2.2 | 1.7 | 1.7 | 1.6 - 2.0 | -21.2 |
| $\rho_{1m}$ (kg m$^{-3}$) | 529 | 591 | 578 | 564 - 597 | 9.3 | 599 | 657 | 646 | 626 - 659 | 7.8 |
| **Georges VI** | | | | | | | | | | |
| ME (Gt yr$^{-1}$) | 15 | 20 | 22 | 16 - 30 | 53.2 | 15 | 20 | 22 | 16 - 30 | 53.1 |
| RU (Gt yr$^{-1}$) | 2 | 3 | 3 | 3 - 4 | 56.9 | 2 | 3 | 3 | 3 - 4 | 56.9 |
| RZ (Gt yr$^{-1}$) | 14 | 18 | 20 | 15 - 27 | 45.8 | 14 | 18 | 20 | 15 - 27 | 45.2 |
| SMB (Gt yr$^{-1}$) | 11 | 10 | 10 | 9 - 11 | -10.3 | 5 | 3 | 3 | 2 - 4 | -25 |
| $LWC_{1m}$ (g kg$^{-1}$) | 2.1 | 1.8 | 2.0 | 1.7 - 2.4 | -4.1 | 3.6 | 3.2 | 3.5 | 2.9 - 4.1 | -4.1 |
| $\rho_{1m}$ (kg m$^{-3}$) | 493 | 537 | 526 | 521 - 537 | 6.8 | 544 | 595 | 584 | 577 - 595 | 7.3 |

preventing the layer under 1m from heating up and latent heat from the refreeze process to be released. Therefore, the snowpack will be cooled down by the underneath layers, and will need more nudging. The second point is that during melt events the upper layers saturate with less water because of the densification. The saturation results in increased runoff and faster percolation of the water into deeper layers, outside of the assimilation depth range. If the model were to retain liquid water in its top snow layers for a longer duration, it would require less nudging to match the RS datasets.

This phenomenon is illustrated in Figure 10, where using a 10 cm assimilation depth threshold for AMSR2 gives more melt production than the 20 cm and the 40 cm threshold, with both the water content threshold at 0.1 % or 0.2 %. The effect ends up being so important that using a 10 cm assimilation depth and 0.2 % $\alpha$ threshold for AMSR2 can result in an intense refreezing and a firn air content depletion that lead to a strong increase in runoff that causes a decrease in SMB for the Antarctic Peninsula. This decrease in SMB is in contradiction with the generally observed trend (Rignot et al., 2019; Chuter et al., 2022).

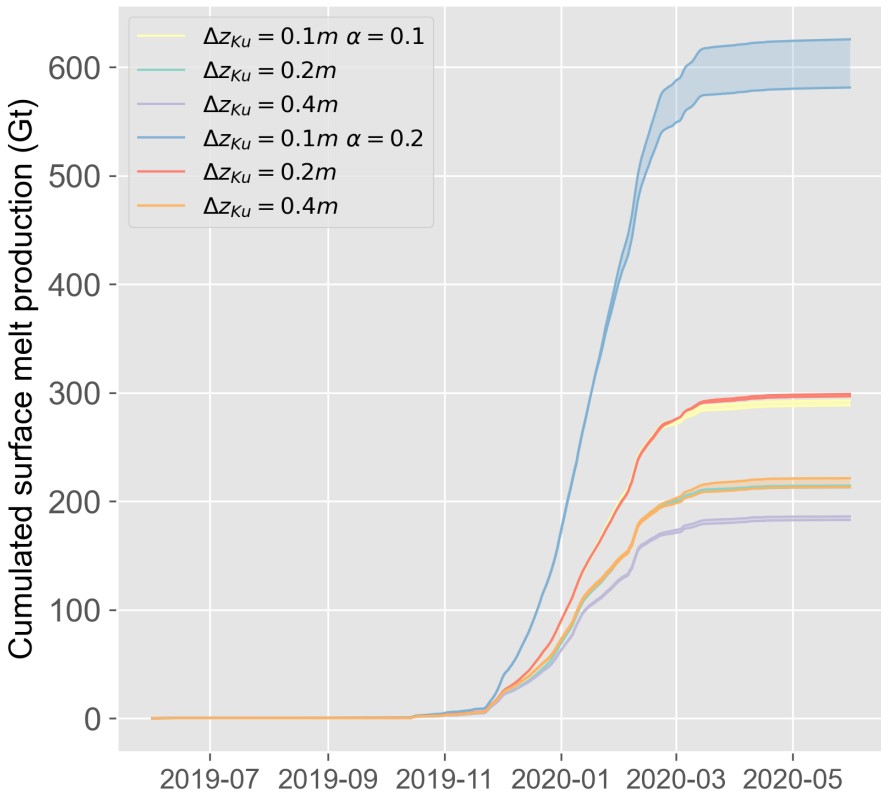

**Figure 10.** Cumulated surface melt production ($\mathrm{Gt}$) for the 2019-2020 melt season as modeled by the different assimilation. The assimilations are grouped by their $\alpha$ and Ku-band $\Delta_z$ thresholds. Shaded areas represent the range of the assimilation of the groups. Groups of assimilations with Ku-band $\Delta_z = 0.1$ m produce more melt than the group of assimilations with the same $\alpha$ but different $\Delta_z$.

Consequently, the three simulations using those parameters for the Ku-band sensors have been discarded to calculate the average melt for the assimilations.

     In contrast, with Sentinel-1, the effect of choosing a different $\Delta_z$ threshold is less pronounced. As shown in Table 4, assimilations that have all parameters in common except the S1 assimilation depth threshold only vary by a few $\mathrm{Gt\,yr^{-1}}$ for all variables. Multiple reasons can explain this comparatively lighter effect. S1 has a larger revisit time compared to AMSR2

(6 days revisit time vs daily images). With fewer images, the specific assimilation depth related to S1 is less frequent in the melt assimilation process within MAR. In addition, as explained previously, the liquid water is kept longer in these slightly deeper layers, and thus no melt is required to reach the water content threshold. The model is thus more sensitive to a shallower assimilation depth threshold. The sensitivity is linked to near-surface events, more likely to occur in the first centimeter of the snowpack. The penetration depth for the C-band sensors is larger than for Ku-band sensors, using sensors with higher

frequencies makes the choice of the thresholds more sensitive.

**Table 7.** Comparison between the number of days between the first day with observed melt and the last one (the melt season length) and the number of melt days modeled for the three studied ice shelves for $MAR_{ref}$ and the average number for assimilations depending on their $\alpha$ between June 2019 and May 2020. A melt day over an ice shelf is considered as a day where more than 10 % of the ice shelf is experiencing melt.

| Larsen C | Melt season length (days) | Number of melt days modeled |
|---|---|---|
| $MAR_{ref}$ | 143 | 90 |
| $\alpha = 0.1\,\%$ | 147 | 110 |
| $\alpha = 0.2\,\%$ | 152 | 119 |

| Wilkins | Melt season length (days) | Number of melt days modeled |
|---|---|---|
| $MAR_{ref}$ | 292 | 127 |
| $\alpha = 0.1\,\%$ | 294 | 125 |
| $\alpha = 0.2\,\%$ | 298 | 129 |

| GeorgeVI | Melt season length (days) | Number of melt days modeled |
|---|---|---|
| $MAR_{ref}$ | 120 | 120 |
| $\alpha = 0.1\,\%$ | 123 | 122 |
| $\alpha = 0.2\,\%$ | 157 | 134 |

### 4.1.2 Water content threshold sensitivity

The water content sensitivity has a smaller impact compared to the assimilation depth. The assimilation influences the number of melt days modeled, thus expanding the melt season duration (Table 7) rather than the quantity of liquid water produced by melting. The required amount of liquid water required to reach the water content thresholds $\alpha$ is small compared to the modeled LWC of a typical melt day. In $MAR_{ref}$, for the 2019-2020 melt season, the value reaches 1.2 % for a melt day on average, above the 0.2 % threshold. For this study, the number of melt days is defined as the number of days of the melt season where 10 % of the ice shelf is experiencing melt, while the melt season length corresponds to the number of days between the first melt day after the first of June and the last melt days before the last day of May of the the following year. Thus, the melt season length also encompasses possible colder periods where no melting event occurs.

Choosing a threshold over the other also influences the average number of melt days on the studied ice shelves (Figure 11). A pixel is considered as melting for the day if the daily-averaged mass of liquid water within the first meter of snow is superior to 0.1 % of the snowpack mass. Therefore, using the 0.2 % threshold over 0.1 % will increase the number of melt days.

By computing the mean value of each pixel number of melt days of the ice shelves, it was found that the largest deviation occurs on Larsen C, with an increase of 15 melt days compared to $MAR_{ref}$. The other two ice shelves exhibit comparatively smaller differences, with Wilkins and George VI experiencing an increase of 8 and 9 melt days, respectively (Figure 11).

Taking the assimilation individually leads to a similar conclusion. The water content threshold choice only emphasizes the differences that are caused by the assimilation depth threshold. It is important to note that the simulations that were discarded

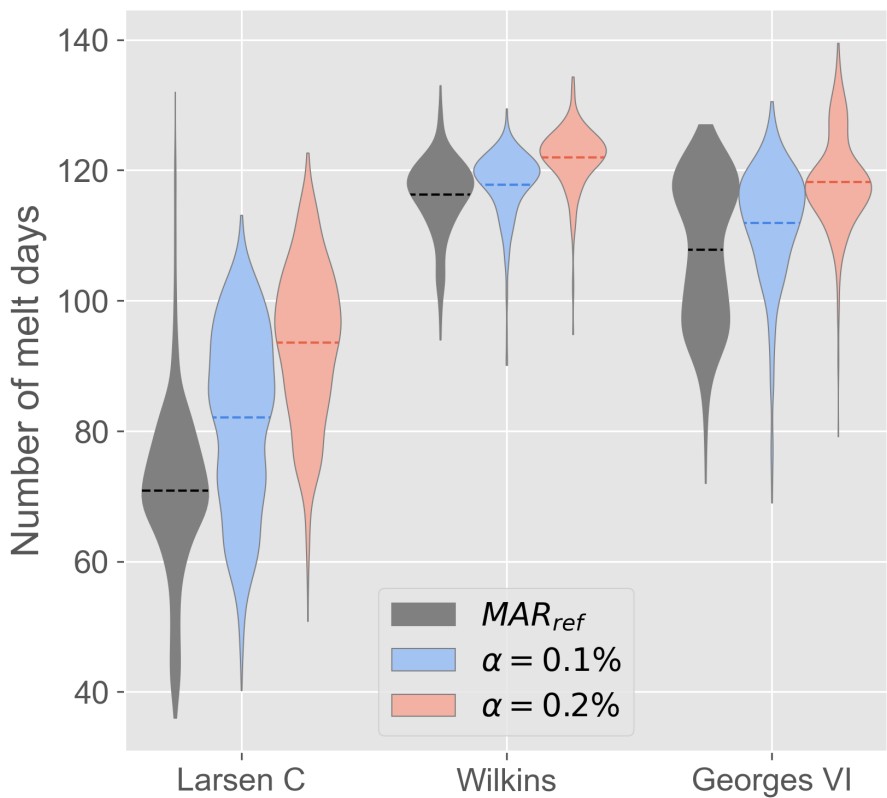

**Figure 11.** Distribution of the number of melt days for the 2019-2020 melt season as modeled by $MAR_{ref}$ and the assimilations grouped by their values of $\alpha$ threshold for the three studied ice shelves. Dashed lines represent the mean value of the distribution. On the three ice shelves, assimilations with $\alpha$ = 0.2 % are experiencing more melt days than $MAR_{ref}$ and the other assimilations. Assimilations with $\alpha$ = 0.2 % are experiencing an increase of the mean number of melt days over the Larsen C ice shelf of 15 days, 8 days on the Wilkins ice shelf, and 9 days on the Georges VI ice shelf.

from the computation of $\overline{Assim}$ are assimilations that had 0.2 % as the value for the threshold. With a densified snowpack, reaching $\alpha$ = 0.2 % required more intense melting.

### 4.1.3 Dataset sensitivity

Each of the four wet-snow masks (AMSR2 desc., AMSR2 asc., ASCAT, S1) has been assimilated individually into MAR to study its influence. Assimilating multiple datasets tend to smooth the sensor characteristics as they are only processed to be used where they provide consistent information. In this study, several characteristics of the remote sensing data have been pinpointed as they influence the results of the assimilation: the acquisition time, the resolution, and the revisit time. They are discussed hereafter.

First, the acquisition time can artificially lower the number of melt days. Because of the daily cycle of the water quantity in the snowpack, images taken earlier in the morning are less likely to observe wet snow (Picard and Fily, 2006). In this manner, over the Antarctic Peninsula, the descending orbit of AMSR2 observes less wet snow than the ascending one. Using satellites whose acquisition times are well distributed during the day allows them to observe the daily melt-refreezing cycle and not miss melt days.

Second, the spatial resolution influences the results of the assimilation because of the pixel heterogeneity. Sensors that have coarser resolution hide a highly heterogeneous surface dynamics and it is possible that while only a fraction of the region covered by one pixel is experiencing melting or enough water is present in the snowpack, the whole pixel is considered as wet snow Picard et al. (2022). In steep regions like near the grounding line, this phenomenon can lead to the detection of wet snow in places where there should not be. In this study, the passive microwave sensor AMSR2 has a coarser resolution than MAR and can trigger the assimilation process where it should not.

To study the influence of the spatial resolution, ASCAT has been assimilated ($AsA_{01}S1_{10}AMA_{02}AS_{02}$ in Table 2) instead of AMSR2 in descending orbit. The assimilations gave smaller numbers of melt days and surface melt production on the Antarctic Peninsula for the studied period (191 $\mathrm{Gt\,y^{-1}}$ for $AsA_{01}S1_{10}AMA_{02}AS_{02}$ and 214 $\mathrm{Gt\,y^{-1}}$ for $Assim_{ref}$). If the assimilation depth is different between AMSR2 and ASCAT, the major influence comes from the spatial resolution of the sensor. The difference can be seen on the wet-snow masks (Figure 12). AMSR2 detects melt on Alexander Island, between George VI and Wilkins ice shelves when ASCAT with a finer resolution and another frequency than AMSR2 does not. Even if wet snow is observed in one of the AMSR masks, the duration of the increased MAR snowpack temperature is too short to produce the water quantities necessary to be detected as a melt day at these places. This preservation of a cold snowpack persists throughout the rest of the day.

Finally, the requirement of the low revisit time is highlighted by studying the wet-snow extent resulting from the assimilation of only one sensor at a time (Figure 13). The S1 wet-snow mask assimilated does not cover the entire AP every day and thus shows a smaller wet-snow extent than the other masks. As a consequence, there are fewer instances in which the model and the mask exhibit discrepancies regarding the snow status, resulting in reduced application of the nudging technique. Eventually, S1-only assimilation ($AsA_{01}S1_{10}$ in Table 2) has the closest wet-snow extent to the extent of $MAR_{ref}$ of the assimilation. The resilience of the model snowpack is such that only relying on a non-daily dataset with intermittent nudging, allows it to freely evolve with minimal external forcing.. Consequently, while the high spatial resolution of Sentinel-1 brings valuable information, this advantage is not sufficient enough to be used as the only dataset assimilated. The S1 dataset needs to be used in conjunction with other datasets to combine high spatial resolution with low revisit time.

The resilience of the snowpack also decreases the feasibility of only assimilating one dataset with the algorithm described in this paper. If ASCAT-only assimilation ($AsA_{01}AS_{10}$ in Table 2) tends to be closest to its wet-snow masks, during peaks of melt (end of November 2019, beginning of 2020) or strong refreeze (mid-March 2020), the effects of nudging do not persist over long time periods and make the required changes of the model to match the observed wet-snow mask.

Assimilating two datasets that entirely cover the studied zone as well as a dataset that has a finer spatial resolution than MAR serves as a means to mitigate the sensitivity of the model to the chosen datasets. The restrained period in which the

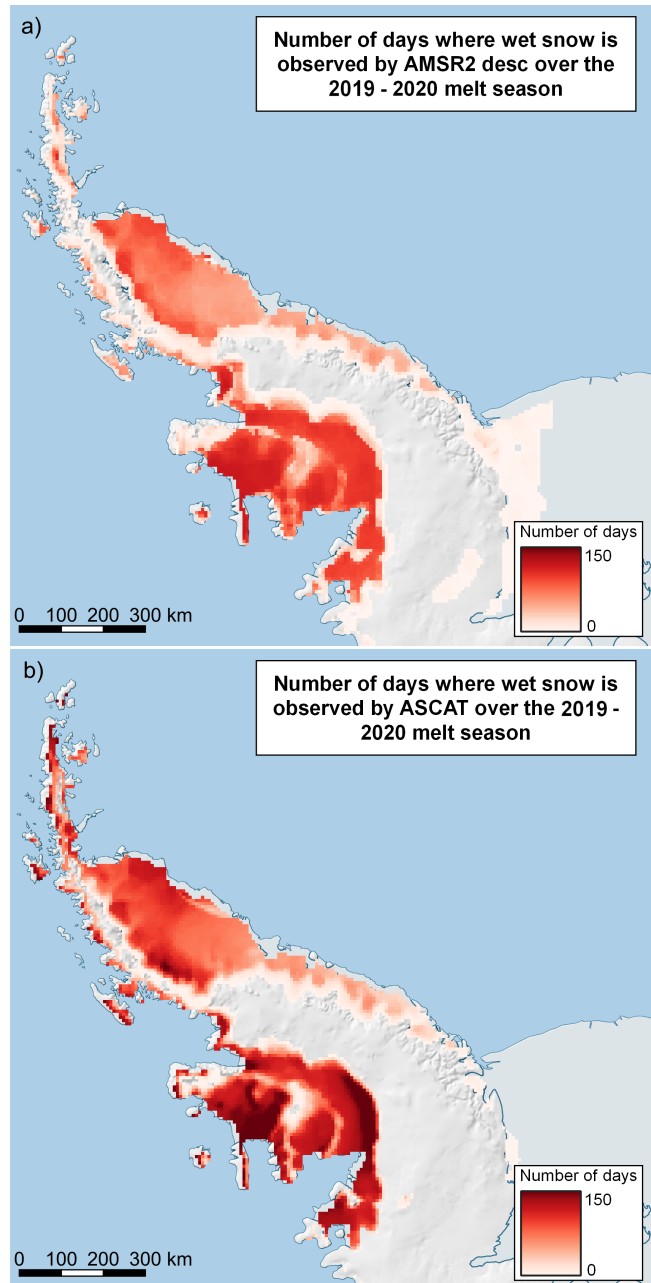

**Figure 12.** (a) Number of days with wet snow observed by AMSR2 ascending on the AP for the 2019-2020 melt season. (b) Number of days with wet snows observed by ASCAT on the Antarctic Peninsula for the 2019-2020 melt season. ASCAT observes more wet snow than AMSR2 over the ice shelves but less in altitude and slopes on average.

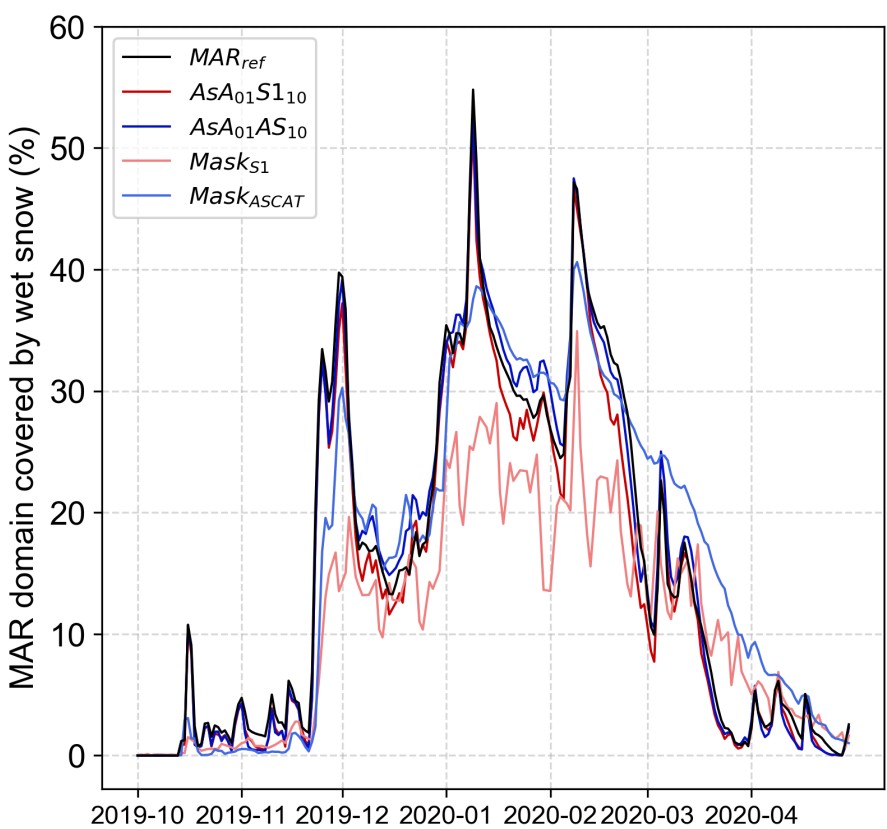

**Figure 13.** Evolution of the wet-snow extent during the 2019-2020 melt season as modeled by $MAR_{ref}$ , the assimilation of only S1 ($AsA_{01}S1_{10}$), the assimilation of only ASCAT ($AsA_{01}AS_{10}$), and the wet-snow masks from S1, and ASCAT. The S1 wet-snow mask has a lower extent as the AP is not covered entirely every day by S1 images.

model snowpack temperature can be changed and the possibility of not assimilating data in case of discrepancy between the sensors also regulate the dependence of the model on the observations. Future developments in the technique should allow the possibility to assimilate more datasets and weighting wet-snow masks according to the relevance of their wet/dry snow status.

## 5   Discussion and conclusion

In this paper, we presented the assimilation of wet-snow occurrence estimated by microwave sensors into the regional climate model MAR. Sensitivity tests have been performed to evaluate the effect of the data assimilation parameters on the model results.

We identified the assimilation depth ($\Delta_z$) to be the most influential parameter when applied for shallow-penetration sensors. The influence on the quantity of water produced in the snowpack partially comes from the liquid water content threshold ($\alpha$)

calculation. The uppermost layer of the snowpack is considerably denser than the underlying layers, owing to the increase in refreezing caused by the exceeding liquid meltwater from assimilation. Heavier and denser layers require more liquid water to reach the required $\alpha$ threshold. Also, the densification causes firn air content depletion, leaving less space for liquid water. The densified layer saturates faster, and more runoff occurs. A threshold of 0.2 m for the Ku-band sensors causes no extreme refreeze or melt and may be considered a good candidate for assimilation depth thresholds. For the C-band sensors, the three thresholds tested yield similar results one or the other, and the implementation of a varying threshold should be considered to take into account the depth at which the wet snow is observed. In contrast to assimilation depth ($\Delta_z$), the LWC threshold ($\alpha$) has a smaller impact on the model surface melt (in Gt). The choice of $\alpha = 0.2$ % over $\alpha = 0.1$ % will mostly increase the duration of the melting season (in number of days).

With constant snowfall (480 GGt y$^{-1}$) and an increase in the surface melt (+95 Gt y$^{-1}$ or +66.7 %), the increase in runoff (+21 Gt y$^{-1}$ or +63.8 %) translates into a decrease in SMB (-4.5 %), for the 2019-2020 melt season. Nonetheless, runoff values are relatively small compared to the surface mass balance, explaining the small impact on the SMB from the assimilation. The general tendency of SMB remains positive in the studied zone. Only the ice shelves show negative SMB during periods of intense melting.

The assimilated dataset was also found to influence the results of the model after data assimilation. Each sensor has its particularities and wet-snow masks may differ from each other. Several of these characteristics have been pinpointed previously. The most important ones are the signal frequency, the revisit time, and the spatial resolution.

The signal frequency of the sensor impacts the resulting melt production by its difference in liquid water sensitivity and the depth to which the signal penetrates. Because it is difficult to provide accurate surface water depth estimates (Fricker et al., 2021) and because microwave signals can be intercepted by the water in the snowpack, the limit at which we stop the assimilation is not always clear. If there is enough water in the top layers, potential liquid water in the deeper layers cannot be observed. In the same way, a thin layer of water can be interpreted as the presence of water in the first meter of the snowpack when the underneath layers are dry (Figure 5). The assimilation depth threshold $\Delta_z$ has been set with different values for the different wavelengths of the sensors but remains constant no matter the wet state of the snowpack. Introducing a LWC/density varying threshold could decrease the melt production after the assimilation. However, we encourage field observation of the evolution of the LWC in the snowpack vertical profile; a required step to introduce and validate the assimilation algorithm.

The revisit time of the satellite is influential as the model freely evolves if the forcing is not performed every day. The assimilation of only Sentinel-1 satellites (revisit time of 6 days, which translates into one image per 2-3 days over the studied zone) is pretty close to the results of the non-assimilated model. Multiple datasets need to be assimilated during the same day for the model to durably change its behavior. The resilience of the model comes from the refreezing of the snowpack during the night and the winter period. When taking into account a few melt seasons, at the beginning of the melt season, the model snowpack is more or less similar to its previous year state.

Assimilating multiple datasets into MAR also brings challenges and consideration alongside its advantages. If some missing information is fulfilled by another dataset, it adds another layer of complexity to the algorithm or additional uncertainties linked to the assimilation method used and its thresholds. Datasets may not carry the same information and may not be compatible

for all the time steps. Here, none of the datasets is considered to have better wet-snow detection than the other. A possible enhancement of the technique would be to add weight to the masks in case of contradiction between them. The weight could be constructed using the confidence level of the wet-snow detection technique employed, the satellite spatial resolution, the topography gradient inside the satellite pixels interpolated to the MAR grid or the sensor sensitivity to water.

The results highlight the importance of data assimilation. While the assimilation does not induce a complete change in the
525 behavior of the model as surface melt remains marginal to snowfall, the snowpack properties tend to deviate from the non-assimilated model impacting in the end the snowpack's ability to retain future meltwater. Here, satellite data have only been assimilated for two melt seasons over a small area. The study can be conducted for a longer period, at a larger scale or over the Greenland ice sheet where surface melt is the main driver of SMB variability (Slater et al., 2021). Further attention should be given to ice shelves as they are the most sensitive region of Antarctica and important to the Antarctic ice sheet stability (Favier
and Pattyn, 2015; Paolo et al., 2015; Sun et al., 2020).

Finally, The results obtained in this paper pinpoint the uncertainties of the regional climate model over the Antarctic Peninsula where, without increasing the snowpack wet extent significantly, the surface melt production significantly increased. The assimilation of remotely sensed data into RCMs is a promising way of reducing the biases and errors inherent to climate models knowing that there is currently no direct large-scale measurement of meltwater content in the snowpack in Antarctica. This is
535 also an easy way to provide robust uncertainties on model outputs over present climate. Using multiple RS datasets with spatial resolution higher than the one of the model will allow correcting the non-assimilated model by better assessing the snowpack water content.

*Code and data availability.* The MAR code used in this study is tagged as v3.12 on https://gitlab.com/Mar-Group/MARv3 (MAR model, 2022). Instructions to download the MAR code are provided on https://www.mar.cnrs.fr (MAR Team, 2022). The MAR outputs used in this
study are available upon request by email (tdethinne@uliege.be). Python code and necessary files to perform the assimilation with MAR are available on https://gitlab.uliege.be/tdethinne/assim_mar

*Author contributions.* TD and XF conceived the study. TD performed the simulations based on a domain of CK. TD led the writing of the manuscript. TD, QG, GP, XF, CK, and AO discussed the results. TD and GP processed the RS data. CK assisted with AWS data comparison. All co-authors revised and contributed to the editing of the manuscript.

*Competing interests.* The authors declare that they have no conflict of interest.

*Acknowledgements.* ERA5 reanalysis data (Hersbach et al., 2020) are provided by the European Centre for Medium-Range Weather Forecasts, from their website at https://www.ecmwf.int/en/forecasts/datasets/reanalysis-datasets/era5 (last access: 24 October 2022).

Consortium des Équipements de Calcul Intensif (CÉCI), funded by the Fonds de la Recherche Scientifique de Belgique (F.R.S. – FNRS) under grant no. 2.5020.11 and the Tier-1 supercomputer (Nic5) of the Fédération Wallonie Bruxelles infrastructure funded by the Walloon Region under grant agreement no. 1117545.

Background maps have been provided by the Norwegian Polar Institute through the Quantarctica3 project (Matsuoka et al., 2021).

This project is carried out in the framework of the Digital Twin Antarctica Project European Space Agency ESA Contract [No.4000128611/19/I-DT].

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
