# Peer review of "Sensitivity of the MAR Regional Climate Model snowpack to the parametrization of the assimilation of satellite-derived wet-snow masks on the Antarctic Peninsula"

_EGUsphere, 2022_

## Referee Comment (RC1)

**1 General comments**

This sensitivity study investigates possible assimilation parameterizations to integrate satellite-derived surface melt extent into regional climate model simulations. Such an assimilation is a promising approach with the potential to improve surface mass balance estimates for Antarctica. Melt extent assimilation could allow to translate satellite-detected melt water into quantitative melt rates and might reduce the uncertainties in model based estimates of surface melt and as such is a much needed development.

The authors use a generally similar assimilation strategy as in Kittel et al. (2022) by nudging the snow temperatures to improve the agreement between simulated and a satellite-derived melt mask. Here, instead of using just one satellite product, the authors include various combinations of data sets which differ in temporal and spatial resolution and penetration depth. Additional parameters tested are the depth down to which modelled snow temperatures are nudged and the minimum in simulated water content which is sufficient to count as an agreement with liquid water detected by satellites.

The paper has a few major issues but also numerous minor, rather technical shortcomings which in my view could have been straightened out beforehand. I recommend major revisions with an more active contribution from experienced co-authors. In some places I have tried to improve the wording, but this is only intended to give an inspiration and makes no claim to be perfect or a complete list.

**2 Major and general comments**

The paper generally identifies different sensitivities for the different assimilation parameters under consideration, however it is not attempted to identify a recommended assimilation algorithm. A basis for such an assessment could be to evaluate which experiment yields good agreement with the binary melt masks with minimal nudging. To this end it could be interesting to compare the values in Table 5 and Figure 11 to respective values diagnosed from the satellite-derived melt masks. Also Figures 12 and 13 might be extended and discussed in greater depth. And it would be interesting to map the total energy added and subtracted (separately) throughout the experiment within each grid box and to provide a budget for the whole domain.

The experiments which use only one data set should also be part of section 2 and should be discussed more systematically and in more detail. Due to the great number of experiments it is difficult to get an overview. It would be helpful to have one table with all experiments (table 2 does not include the one sensor experiments) and another one with some simple metrics such as the number of melt days and the total meltwater production over the whole melt season for each experiment.

Liquid water in the snow pack is not necessarily indicating ongoing melting- it can also indicate past melt events with incomplete refreezing at night (e.g. for cloudy conditions or at greater depth)- this should be distinguished and also discussed with respect to the different duration of the melt season for the different satellite data sets. Also I would not use the term of "binary melt masks" - but something like "wet snow masks". Furthermore I think that possibly valuable information is discarded when shallow penetrating data sets indicate no wet snow while the deep penetrating data set indicates wet snow. This is not necessarily a conflict but could occur after a melt period has ended but percolated water may remain liquid in deeper layers.

As the satellites only detect presence of liquid water and not melt, I am also suprised that sensor penetration depth and assimilation depth are so closely linked here: I would not expect melt at depths of 1 m or more. I would rather have limited

the temperature nudging to a much shallower surface layer. However one could still compare water content (here I would use absolute and not relative values) down to the respective penetration depths of the available measurments and then trigger melt only in the surface layer. I wonder if there are reasons against such a strategy.

5    Sometimes the word *sensitivity* seems to be used ambiguously. Most of the time it is used as in "simulated melt (whether more or less) depends strongly on the parameter choice" but for instance in l.14-15 it seems to rather mean "more melt is detected for a certain parameter choice"

10    The words used for the assimilation parameters are unnecessary diverse and confusing. I would recommend to consistently use something like melt water threshold and assimilation depth. The latter should not be named threshold in my view and it also should not be called penetration depth as this can be easily confused with the penetration depth of the individual sensors.

15    The introduction is too unstructured, I give some specific comments below, but these should be only considered after sorting the different aspects in a linear fashion.

Also the method part is hard to read and should be thoroughly revised.

20    Maybe it is not a problem for people from the remote sensing community- but the paper is not easily readable for the wider community. For instance, datasets are sometimes referred to by their mission (Sentinel), the general measurement (radar, radiometer, scatterometer), some general classifications (active or passive sensors) or their instrument name (ASCAT)- this is unnecessarily confusing.

**3    Some specific comments**

25    Title: it is the MAR snowpack which is sensitive to the assimilation, not the satellite-derived surface melt. Maybe: Assimilation of satellite-derived surface melt into the regional climate model MAR: sensitivity of the snowpack on the Antarctic peninsula to assimilation parameters

Abstract:
30    l. 1: please reword the whole sentence and possibly add 1 or 2 sentences. Here it would be good to introduce the problem (e.g. surface melt, runoff and accumulations cannot be directly observed on larger scales and models have uncertainties, remote sensing can only provide melt extent)
l. 3: rather use "reduce uncertainties"
l. 18: maybe: second parameter mostly influences the duration of the melt period but it has only limited effect on the absolute
35    melt water production.

Introduction:
l.29: maybe: even moderate surface melt is thought to weaken ... leading to substantial mass loss.
ll. 31-34 too long, muddled, partly redundant.
40    l. 35 climate models do not monitor (wrong verb), they do not comprise the ice body and only few include the snow pack.
l. 47: correct: induce -> induced
l. 49: delete: In addition
ll. 54-61: this is better placed earlier in the section and merged with the earlier sentence about ice shelves
l. 62: be specific, here: melt -> melt areas
45    l. 62: include Kittel et al. (2022) here and generally explain that the strategy is to warm or cool the snow pack in order to better match satellite derived melt maps.

ll. 66-72: this is too general and does not get to the point. I think you wanted to say that the different available products yield either poor spatial or temporal resolution and in contrast to Kittel et al. (2022) you test combining several products.

l. 68: active / passive sensors should be explained in this journal before using these terms

5   Section 2.1: this section could be better structured. Consider implementing subsections for each satellite/sensor type and providing a table with technical specifications (e.g. mission, sensor, resolution, revisit time, reference). It would be good to have a table with unique names for the four data sets and their technical specifications, and then to only use the data set names.

l. 78: four data sets from three sources?

l. 78: the fact that the strategy is to produce binary melt masks from satellite data and assimilate these should already be spelled
10  out in the introduction and abstract.

l. 90: this is confusing as here only three data sets are mentioned

l. 94: "level 3 products" seems to be an unnecessary detail.

l. 96: ascending and descending paths should be explained.

eq. 10: how is TP measured?

15  l. 104: the reference to Fig. 3 is confusing here; I propose to refer to Fig. 3 in l.99 and refer to Fig. 2 at the end of l. 105.

l. 116: maybe better: sensors will indicate the presence of water .. by changes in the backscattering.

l. 125: this is not coming to the point: the -2.66 dB threshold is used in this study?

p. 7, Fig. 3: use coast line contours also in the upper panels

l. 140: explain or avoid the word scene in this context

20  l. 145: it is unclear to which part the word "else" is related to.

p. 8, Fig 4: it could be interesting to see panel B after normalization.

l. 146: A figure for the ASCAT data could be included, similar to Figs. 3 and 4. Also a reference for this data set is missing.

Section 2.2:

25  l. 157: "transfer between atmospheric part ... and the atmosphere" is this right?

l. 159: What is the typical vertical resolution in the upper 1.5m?

Also please cover the percolation algorithm which seems to be crucial to understand the response in subsurface liquid water content

30  Section 2.3:

Table 2 should be introduced in this section. Also it should include the single-data set experiments and I find experiment name $MAR_{a01-ku-02-c10}$ unfortunate as it does not indicate that here a different input is used and it does not indicate the assimilation depth of the third data set. Generally the experiment names are not very handy. I would suggest something like $AsSd_l$ for AMSR with shallow assimizlation depth+S1 with deep assimizlation depth and low water content threshold.

35  l. 173-174: check grammar

l. 175: correct: As up to three...

l. 183: is it possible to heat beyond $0^oC$?

l. 187: either percolate into the ice or accumulate in the ice

l. 190: better: discarded -> ignored

40  l. 192: shorter: if the two masks agree, the two observations...

l. 199: more precise: at the same time -> within the same 3-hour time window

l. 206-207: maybe put this first

l. 213: check unit

45  Section 3:

Since the assimilation and the analysis are dealing with the snowpack it would be helpful to also evaluate precipitation and melt. Maybe compare to Wang et al. (2021).

l. 239: correlation **(r)**

l. 249: better: a weak correlation and/or a strong negative bias

l. 250: actually the bias is also strongest in summer (winter insolation should be weak anyway) and biases in net longwave radiation and net shortwave radiation almost cancel out and indicate underestimated cloud cover.

l. 255: "Combined with ...": unclear

5 Section 4:

$Assim_{mean}$ is an unfortunate name, as it sugests an experimemt of its own right- I would propose $\overline{Assim}$ or mean(Assim). Also it should be stated here that three experiments were discarded in $Assim_{mean}$

l. 258: correct is->are

l.: 266: correct model-> simulation

10 l. 269: gives different results from -> differs from

Table 3: Evolution is not (but should be) mentioned in the caption- I understand that evolution is relative change due to assimilation in $Assim_{mean}$, the name evolution is maybe misleading. It is not clear whether LWC and $\rho$ are mean state or final states at the end of the period. Another column for $LWC_5m$ or some other deeper layer would be interesting. Also: replace *mean value of the assimilations* with *mean value of the 16 assimilations selected for $Assim_{mean}$*

15 Figure 7: runoff could be shown in the same figure with a different y-axis on the right, maybe also highlight $Assim_{ref}$

l. 271: rephrase without the first part

l. 275: please check: 63.8 is the value for runoff according to table 3.

l. 279: the same -> almost the same

l. 280 ff: clumsy, please rephrase.

20 l. 282: correct: depending on the energy balance

l. 284: densify -> densifies

l. 287: SMB is either $snowfall + winddrift + refreeze - melt - sublimation$

or $precipitation + winddrift - runoff - sublimation$

also please clarify whether snow drift is represented in MAR.

25 l. 292: please specify "deeper"

l. 293: not all ice shelves exhibit lower liquid water quantity

l. 295ff: Table 4 should be discussed in more detail: there is no explanation given, why LWC for Larsen C is increasing. Also for individual ice shelves it is not true that relative changes in melt and runoff are of similar size. It is particularly not true for Wilkins where additonal melt almost entirely becomes additional runoff. For a better understanding it could help to map the

30 degree of saturation in the upper snow pack and to look at different stages of the melt season. A deeper interpretation of Table 3 and 4 is also difficult due to ambiguous variable definitions (see below)

ll. 299-302: this is completely unclear to me.

l. 303: It is not really suprising that the mean of the assimilation experiments is close to the central reference experiment. However without evaluation this is not necessarily meaning that this is more realistic than other members.

35 Table 4: The caption to this table is sloppily formulated. As in table 3 not all lines and columns are well defined or self explanatory and there is room for guesswork but no sound basis for interpretation. Also I wonder why LWC is consistently one order of magnitude smaller than in Table 3.

l. 310: refreezing is indeed releasing energy and heating the ambient snow! Please revise ll. 310-314.

l. 310: not sure what prevails means here. Maybe prevail->prevent?

40 l. 317: please specify what exactly qualifies results to be inprobable. Such a exclusion criterion should be defined beforehand. And the exclusion of the members should be stressed when introducing table 2 in the method section.

l. 333: lesser -> less or smaller

Table 5: Which are the experiments considered here? Are these numbers the same for all experiments with $\alpha = 0.1$ and $\alpha = 0.2$? Also maybe noteworthy: number of melt days larger for $MAR_{ref}$ than one of the assimilations on Wilkins.

45 l. 343: is this referring to the whole 20m snow pack?

l. 347: I don't find these numbers in Table 5- I stop reading this paragraph here.

l. 364: this cycle -> daily melt - refreezing cycle?

l. 373: is other frequency here higher frequency?

section 5: this probably needs to be rewritten anyway after the other parts of the manuscript have been revised.

l. 394: effect of assimilation was not studied here.

**References**

Kittel, C., Fettweis, X., Picard, G., and Gourmelen, N.: Asimilation of satellite-derived melt extent increases melt simulated by MAR over the Amundsen sector (West Antarctica), Bulletin de la Société Géographique de Liège, 78, 87–99, 2022.

Wang, Y., Ding, M., Reijmer, C. H., Smeets, P. C. J. P., Hou, S., and Xiao, C.: The AntSMB dataset: a comprehensive compilation of surface mass balance field observations over the Antarctic Ice Sheet, Earth System Science Data, 13, 3057–3074, https://doi.org/10.5194/essd-13-3057-2021, https://essd.copernicus.org/articles/13/3057/2021/, 2021.

---

## Author Comment (AC1)

Reviewer #1

We would like to thank the reviewer #1 for its constructive comments that will improve the paper. We have responded to all of them and will modify our paper accordingly. The majority of the highlighted issues of the paper come from its writing style. The paper will undergo a restructuring and will be sent to a native english speaker after rewording to reduce confusions. Our point-by-point answers (in bleu) follow as a supplement.

Major and general comments:

*The paper generally identifies different sensitivities for the different assimilation parameters under consideration, however it is not attempted to identify a recommended assimilation algorithm. A basis for such an assessment could be to evaluate which experiment yields good agreement with the binary melt masks with minimal nudging. To this end it could be interesting to compare the values in Table 5 and Figure 11 to respective values diagnosed from the satellite-derived melt masks. Also Figures 12 and 13 might be extended and discussed in greater depth. And it would be interesting to map the total energy added and subtracted (separately) throughout the experiment within each grid box and to provide a budget for the whole domain.*

> An attempt to identify the optimal parameters will be performed by analyzing the difference between the satellite products and the modeled number of melt days. For the energy balance, the exact value can not be inferred from the results as simulations need to be performed again for having these outputs. However, an estimation can be obtained by calculating the quantity of snow melted/refrozen compared to the reference simulation and multiplying it by the energy required to change snow/water state.

*The experiments which use only one dataset should also be part of section 2 and should be discussed more systematically and in more detail. Due to the great number of experiments it is difficult to get an overview. It would be helpful to have one table with all experiments (table 2 does not include the one sensor experiments) and another one with some simple metrics such as the number of melt days and the total meltwater production over the whole melt season for each experiment.*

> Table 2 will be extended to include one-sensor experiments. Another table will be added to give an overview of the results of the different experiments.

*Liquid water in the snow pack is not necessarily indicating ongoing melting- it can also indicate past melt events with incomplete refreezing at night (e.g. for cloudy conditions or at greater depth)- this should be distinguished and also discussed with respect to the different duration of the melt season for the different satellite data sets. Also I would not use the term of "binary melt masks" - but something like "wet snow masks". Furthermore I think that*

*possibly valuable information is discarded when shallow penetrating data sets indicate no wet snow while the deep penetrating data set indicates wet snow. This is not necessarily a conflict but could occur after a melt period has ended but percolated water may remain liquid in deeper layers.*

As satellites observe liquid water in the snowpack and not the melting phenomenon itself, using "*Binary melt masks*" as we did is indeed misleading and will therefore be changed to wet-snow masks. A similar change will also be performed for all the mention of a sensor "*detecting melt*". We discarded the satellite observations when shallow-penetration sensors do not detect liquid water but a deeper penetrating sensor does in a manner of consistency. This difference in detection can come from deeper water that is not observed but could also be caused by a difference in the sensitivity of the sensor or even a false detection. This is correct that valuable information is lost, but as we wanted to keep a balance between the complexity of the algorithm and computation speed, these cases are currently not taken into account.

*As the satellites only detect presence of liquid water and not melt, I am also surprised that sensor penetration depth and assimilation depth are so closely linked here: I would not expect melt at depths of 1 m or more. I would rather have limited 1 the temperature nudging to a much shallower surface layer. However one could still compare water content (here I would use absolute and not relative values) down to the respective penetration depths of the available measurements and then trigger melt only in the surface layer. I wonder if there are reasons against such a strategy.*

The assimilation algorithm is based on Kittel et al. (2022) which nudges the snowpack temperature to the satellite penetration depth. As we do not have information on the depth of liquid water and its amount with the satellite imagery and the sensors are very sensitive to the presence of liquid water, the default warming depth is set equal to the penetration capabilities of the sensor. MAR can hold water in one layer until 5% of the air bubbles contained in the snow are filled with water before making it percolate to the underneath layer. By heating the snowpack uniformly, it is possible to have a more uniform presence of water.

*Sometimes the word sensitivity seems to be used ambiguously. Most of the time it is used as in "simulated melt (whether more or less) depends strongly on the parameter choice" but for instance in l.14-15 it seems to rather mean "more melt is detected for a certain parameter choice"*

As this wording is confusing, we will revise it at each occurrence and change it accordingly in the revised version.

*The words used for the assimilation parameters are unnecessarily diverse and confusing. I would recommend to consistently use something like melt water threshold and assimilation depth. The latter should not be named threshold in my view and it also should not be called penetration depth as this can be easily confused with the penetration depth of the individual sensors*

> The name of the threshold will be uniformized throughout the paper and the name *"penetration depth"* will be changed to *"assimilation depth"* when it does not specifically refer to the penetration depth of the satellite.

*The introduction is too unstructured, I give some specific comments below, but these should be only considered after sorting the different aspects in a linear fashion.*

> The structure of the paper will be revised as well as the structure of the different parts of the paper. A strong revision by the co-authors and English native speakers will be performed.

*Also the method part is hard to read and should be thoroughly revised*

> As for the introduction, the methodology part will be revised to clarify and simplify to facilitate the understanding of this part. In this manner, multiple changes will be applied :
> - Subsections describing each satellite dataset will be added.
> - The description of the assimilation algorithm will be extended to follow Figure 5.
> - The ll. 205-207 will be moved before the assimilation cases explanations.
> - Subsections about the parameters will be corrected following the minor comments.

*Maybe it is not a problem for people from the remote sensing community- but the paper is not easily readable for the wider community. For instance, datasets are sometimes referred to by their mission (Sentinel), the general measurement (radar, radiometer, scatterometer), some general classifications (active or passive sensors) or their instrument name (ASCAT)- this is unnecessarily confusing.*

> It is true that in remote sensing, some dataset are referred to as the name of the mission or the platform rather than the name of the sensor (mainly when there is only one sensor aboard the platform, like Sentinel-1 or Sentinel-2). These specificities will be presented when the datasets are described, then datasets will more simply be referred to by the name of the satellite (i.e. *"Sentinel-1 (S1)"*, *"AMSR2"* and *"ASCAT"*) in the revised version to decrease the confusion it caused.

Specific comments:

*Title: it is the MAR snowpack which is sensitive to the assimilation, not the satellite-derived surface melt. Maybe: Assimilation of satellite-derived surface melt into the regional climate*

*model MAR: sensitivity of the snowpack on the Antarctic peninsula to assimilation parameters.*

> *The title will be changed to: "Sensitivity of the Regional Climate Model MAR's snowpack to the assimilation parametrization of satellite-derived surface wet snow on the Antarctic Peninsula".*

*l. 1: please reword the whole sentence and possibly add 1 or 2 sentences. Here it would be good to introduce the problem (e.g. surface melt, runoff and accumulations cannot be directly observed on larger scales and models have uncertainties, remote sensing can only provide melt extent).*

> Abstract will be reworded and the problematics related to *in situ* observations and remote sensing will be included in the revised version.

*l. 3: rather use "reduce uncertainties".*

> Noted and it will be changed accordingly in the revised version.

*l. 18: maybe: second parameter mostly influences the duration of the melt period but it has only limited effect on the absolute melt water production.*

> This sentence will be reworded to: *"For the second threshold, the impact is more important on the number of melt days [days] rather than the melt production [Gt] itself".*

*l.29: maybe: even moderate surface melt is thought to weaken ... leading to substantial mass loss.*

> Noted and it will be changed accordingly in the revised version.

*ll. 31-34 too long, muddled, partly redundant.*

> This part will be removed in the revised version.

*l. 35 climate models do not monitor (wrong verb), they do not comprise the ice body and only few include the snow pack.*

> Noted and it will be changed accordingly in the revised version.

*l. 47: correct: induce -> induced.*

> Noted and it will be changed accordingly in the revised version.

*l. 49: delete: In addition.*

> Noted and it will be deleted in the revised version.

*ll. 54-61: this is better placed earlier in the section and merged with the earlier sentence about ice shelves.*

> The sentence will be included earlier as the structure of the introduction will be changed.

*l. 62: be specific, here: melt -> melt areas.*

> Noted and it will be changed accordingly in the revised version.

*l. 62: include Kittel et al. (2022) here and generally explain that the strategy is to warm or cool the snow pack in order to better match satellite derived melt maps.*

> Kittel et al., (2022a) will be added in the introduction and the general concept of its assimilation strategy will be briefly explained here.

*ll. 66-72: this is too general and does not get to the point. I think you wanted to say that the different available products yield either poor spatial or temporal resolution and in contrast to Kittel et al. (2022) you test combining several products.*

> These lines are supposed to explain the main advantages/disadvantages of the passive/active sensor and that both can be used together to benefit from the advantages of the two types of sensors. As this is currently not clear, the part will be entirely rewritten.

*l. 68: active / passive sensors should be explained in this journal before using these terms.*

> An explanation of passive/active sensors will be added at the same time as the subsection describing the remote sensing datasets in the revised version of our manuscript.

*Section 2.1: this section could be better structured. Consider implementing subsections for each satellite/sensor type and providing a table with technical specifications (e.g. mission, sensor, resolution, revisit time, reference). It would be good to have a table with unique names for the four data sets and their technical specifications, and then to only use the data set names.*

> Subsections dedicated to each remote sensing datasets will be added as well as a table taking into account their specification to reduce the confusion induced by the name of the datasets.

*l. 78: four data sets from three sources?*

> The AMSR2 data are splitted in two datasets : one with only ascending mode acquisitions and one with descending mode acquisition. *In fine* we have the

Sentinel-1 dataset, ASCAT dataset, AMSR2 ascending dataset, AMSR2 descending dataset.

*l. 78: the fact that the strategy is to produce binary melt masks from satellite data and assimilate these should already be spelled out in the introduction and abstract.*

It will be stated in the introduction and abstract that we produce **wet snow masks** we want to assimilate.

*l. 90: this is confusing as here only three data sets are mentioned.*

It will be clarified in the revised version as explained for the comments on line 78.

*l. 94: "level 3 products" seems to be an unnecessary detail.*

The level of a product in remote sensing stands for the operations that are already applied to the product before downloading it. Here, we specify the level as we did not calculate the temperature brightness myself. It also answers the question: "How does the author computed the temperature brightness?". Here we will simply specify the dataset we use by citing the location it can be retrieved.

*l. 96: ascending and descending paths should be explained.*

Ascending and descending will be explained before, in the AMSR2 subsection, following l. 78 comment.

*eq. 10: how is TP measured?*

We did not calculate TP as we use the provided level-3 product that contains the temperature brightness, as mentioned in l. 94 response to comment. Eq1 is there to remind us that the temperature brightness of a body will change if its emissivity changes but not its physical temperature.

*l. 104: the reference to Fig. 3 is confusing here; I propose to refer to Fig. 3 in l.99 and refer to Fig. 2 at the end of l. 105.*

Comments from anonymous reviewers #1 and #2 tend to suggest opposite directions (either simplify, or on the contrary go in more detail). Consequently, the reference to the figures may change. Considering the viewpoint of the second reviewer, having 3 figures (Fig. 2,Fig. 3, and Fig. 4) on melt detection, which is not the main subject of this paper, may be redundant. Fig 3 will be removed as well as its reference line 104.

*l. 116: maybe better: sensors will indicate the presence of water .. by changes in the backscattering.*

This sentence will be reworded to : *"It is possible to detect the presence of liquid water in the snowpack in Sentinel-1 images by identifying changes in backscattering coefficient σ0 through time (Figure 4)"*.

*l. 125: this is not coming to the point: the -2.66 dB threshold is used in this study?*

Yes the -2.66 dB is used on the normalized images of Sentinel-1 as threshold to detect wet snow.

*p. 7, Fig. 3: use coastline contours also in the upper panels.*

Coastline will be added in Figures that do not include them for better readability.

*l. 140: explain or avoid the word scene in this context.*

Noted, the word *"scene"* will be avoided.

*l. 145: it is unclear to which part the word "else" is related to.*

Sentences will be reworded . Sentences from ll.143-145 will be reworded to : "

*To create daily wet-snow masks, Sentinel-1 images of the same day were combined. In the case where three or more images overlap, the snow state is selected by a majority filtera and the acquisition time is defined as the mean time between the selected acquisitions. In the case where there are only two images and that contradict each other, the non-wet status is assumed. The acquisition time selected is then the acquisition time of the non-wet image."*

*p. 8, Fig 4: it could be interesting to see panel B after normalization.*

It will be added in the revised version.

*l. 146: A figure for the ASCAT data could be included, similar to Figs. 3 and 4. Also a reference for this data set is missing.*

As explained previously, as the melt detection is not the main subject of the paper, adding another figure may complexify the paper although the ASCAT description will be extended. ASCAT dataset can be retrieved from the EUMETSAT data hub (https://data.eumetsat.int/data/map/EO:EUM:DAT:METOP:ASCSZF1B).

*l. 157: "transfer between atmospheric part ... and the atmosphere" is this right?*

It is a typo, as it should be *"transfer between atmosphere and soil"*.

*l. 159: What is the typical vertical resolution in the upper 1.5m? Also please cover the percolation algorithm which seems to be crucial to understand the response in subsurface liquid water content.*

MAR is configured with a decreasing vertical resolution of the layers from the top to the bottom. The fists layers are typically at the centimeter size while under the first meter, we are at the meter resolution. The four first maximum layer thickness are respectively 2, 5, 10, 30 cm for example.

The percolation algorithm of MAR will be explained in more detail in the subsection 2.2 dedicated to model description. Here we worked with a maximum of the total layer weight being composed at 5% of liquid water before starting percolation.

*Table 2 should be introduced in this section. Also it should include the single-data set experiments and I find experiment name MARa01−ku−02−c10 unfortunate as it does not indicate that here a different input is used and it does not indicate the assimilation depth of the third data set. Generally the experiment names are not very handy. I would suggest something like AsSdl for AMSR with shallow assimilation depth+S1 with deep assimilation depth and low water content threshold.*

Table 2: Table 2 will be extended and placed in the introduction. The naming convention of the experiments have been changed and will be described in text to include the name of the sensor included for the assimilation. *E.g.* MARa01−ku02−c05 became $AsA_{01}S1_{05}AMA_{02}AMD_{02}$.

*l. 173-174: check grammar.*

Noted and changed accordingly in the revised version.

*l. 175: correct: As up to three...*

Noted. It will be corrected.

*l. 183: is it possible to heat beyond 0 °C?*

No, the snowpack temperature can not be heated beyond 0°C. Snow can not have a temperature higher than 0°C. At this temperature, snow is transformed into liquid water if more energy is available. Also, snowpack can not be cooled down under -7.5°C in our algorithm.

*l. 187: either percolate into the ice or accumulate in the ice.*

Noted and it will be changed to "accumulate in the ice".

*l. 190: better: discarded -> ignored.*

Noted. It will be corrected in the revised version.

*l. 192: shorter: if the two masks agree, the two observations . . .*

Noted and it will be changed to *"If the two masks agree, the two observations are associated with the first case …"* in the revised version of the manuscript.

*l. 199: more precise: at the same time -> within the same 3-hour time window.*
Noted and iIt will be corrected accordingly in the revised version of the manuscript.

*l. 206-207: maybe put this first.*
Lines will be moved before the explanation of the different cases.

*l. 213: check unit.*
It should be Kg/m².

*Section 3: Since the assimilation and the analysis are dealing with the snowpack it would be helpful to also evaluate precipitation and melt. Maybe compare to Wang et al. (2021).*
As also pinpointed by the other reviewer, the MAR evaluation will be explained in further detail in the revised version and the dataset with which the evaluation is performed will be better stipulated. First, the observation data provided in Wang et al. (2021) does not cover the zone/studied period which makes the comparison complex. Second, the surface melt can also be evaluated by comparing the results with Jakobs et al. (2020). Results of this evaluation still need to be made carefully. Jakobs dataset also remains a modeled-based estimate, with its own biases and limitations, and therefore cannot be used as if there were in situ measurements, and so a reference for the model estimations. However, a short comparison of $MAR_{ref}$ with the AWS used by Jakobs available on The Antarctic Peninsula can still be added to the paper.

Although, as we are testing the sensibility of the model, we only evaluate MAR without assimilation as we know that the value given after assimilation will differ from the observations.

As represented in the Figure here under (Figure R1.1) we compare the melt estimates from the AWS described in Jakobs et al. (2020). to the surface melt production of the 4 closest MAR pixels of the AWS. MAR has a tendency to overestimate some extremes of melting while simultaneously underestimating or overestimating the duration of periods during which the ice shelves are experiencing melting. Even if it is important to note that there can be a difference in altitude between the AWS and MAR pixels that explains the differences between the two datasets, this comparison also highlights the importance of nudging MAR to correspond to the remote sensing observation of wet snowpack.

[Figure]

*Figure R1.1 : Comparison of surface melt production as modeled by MAR without assimilation and estimated surface melt production from AWS 14, 15, 17, and 18 described in Jakobs et al. (2020).*

*l. 239: correlation (r).*

Noted. It will be added in the revised version.

*l. 249: better: a weak correlation and/or a strong negative bias.*

Noted and it will be changed to "*A weaker correlation is observed in summer for long-wavelength downward radiations (r = 0.65).*" in the revised version.

*l. 250: actually the bias is also strongest in summer (winter insolation should be weak anyway) and biases in net longwave radiation and net shortwave radiation almost cancel out and indicate underestimated cloud cover.*

The biases are indeed caused by the cloud cover. The effect of clouds will be explained in more detail. The explanations of the effect of cloud cover in the simulation will be based on Kittel et al. (2022).

*l. 255: "Combined with ...": unclear*

It will be changed to: "*The implementation of cloud microphysics and the radiative scheme implemented (Kittel 2022b) suggest that MAR underestimates the liquid water path during summer when compared to Cloudsat-CALISPO estimates. Such underestimation is partially responsible for the LWD bias observed in summer.*"

*Assimmean is an unfortunate name, as it suggests an experiment of its own right- I would propose Assim or mean(Assim). Also it should be stated here that three experiments were discarded in Assimmean*

Name of the experiments will be changed in the revised version. $\text{Assim}_{mean}$ will be renamed to $\overline{Assim}$.

*l. 258: correct is->are*

Noted. It will be corrected in the revised version.

*l.: 266: correct model-> simulation*

Noted. It will be corrected in the revised version.

*l. 269: gives different results from -> differs from*

Noted. It will be corrected in the revised version.

*Table 3: Evolution is not (but should be) mentioned in the caption- I understand that evolution is relative change due to assimilation in Assimmean, the name evolution is maybe misleading. It is not clear whether LWC and ρ are mean state or final states at the end of the period. Another column for LW C5m or some other deeper layer would be interesting. Also: replace mean value of the assimilations with mean value of the 16 assimilations selected for Assimmean*

Table 3 will be changed to include mean value during summer and winter. Caption will be changed to : "*Change of surface melt production (ME), runoff (RU), surface mass balance (SMB), snowpack density (ρ), and snowpack liquid water content (LWC) for $MAR_{ref}$ and the mean value of the assimilations ($\overline{Assim}$) over the Antarctic Peninsula. Variables are cumulated over summer (DJF) and winter (JJA) except for snowpack density which is the average density for the seasons.*
*LWC and ρ are given for the 20 firsts centimeters and the firsts meter of the snowpack while the other variables are given as for the whole modeled snowpack.*"
As most of the variables remain constant or null during winter, the table discussion should not be different.

*Figure 7: runoff could be shown in the same figure with a different y-axis on the right, maybe also highlight Assimref*

Runoff and refreeze will be included in the figure.

*l. 271: rephrase without the first part*

"*Although there are divergences while using different parameters in the assimilation,*" will be removed in the revised version.

*l. 275: please check: 63.8 is the value for runoff according to table 3.*

It should have been 66.7%. Noted. It will be corrected.

*l. 279: the same -> almost the same*

Noted. It will be changed to "*almost similar*".

*l. 280 ff: clumsy, please rephrase.*

l279 - 281 will be reworded to "Despite the fact that the relative increase in surface melt and runoff is almost similar for $\overline{Assim}$ and MAR$_{ref}$ (66.7% and 63.8%, respectively), their absolute increase in Gt y-1 is not the same (+95 Gt y−1 and +21 Gt y−1, respectively). This suggests that the snowpack can still absorb liquid water unless it reaches its maximum capacity."

*l. 282: correct: depending on the energy balance*

Noted. It will be corrected.

*l. 284: densify -> densifies*

Noted. It will be corrected.

*l. 287: SMB is either snowfall + windrift + refreeze − melt − sublimation or precipitation + winddrift − runoff − sublimation also please clarify whether snow drift is represented in MAR.*

In the case of this study, SMB= snowfall + rainfall - sublimation - runoff as the blowing snow module was not active for these simulations.
Blowing snow module was turned off to increase simulation speed and to not impact the sensitivity.

*l. 292: please specify "deeper".*

The word "*deeper*" will be removed as it brings nothing to the discussion.

*l. 293: not all ice shelves exhibit lower liquid water quantity.*

l. 293 will be reworded to " *With a denser snowpack, firn air content is reduced and there is less space for liquid water to be absorbed. Therefore, despite the increase in surface melt production, the assimilation process may lead to a decrease in the amount of liquid water retained in the snowpack. This is because the assimilation causes a reduction in the snowpack's capacity to retain water.*"

l.295ff: *Table 4 should be discussed in more detail: there is no explanation given, why LWC for Larsen C is increasing. Also for individual ice shelves it is not true that relative changes in melt and runoff are of similar size. It is particularly not true for Wilkins where additional melt almost entirely becomes additional runoff. For a better understanding it could help to map the degree of saturation in the upper snowpack and to look at different stages of the melt season. A deeper interpretation of Table 3 and 4 is also difficult due to ambiguous variable definitions (see below)*

l 295 will be reworded to : "*All three highlighted ice shelves (Larsen C, Wilkins, and Georges VI) are expericening a increase in surface melt production and runoff (Table 4). Similar conclusion drawn for the AP can also be applied for them.*"

The evolution of LWC will be studied in more detail as some bias induced by the nudging of the snowpack temperature could be overlooked by only using this table. The data presented in Figure R1.2 indicates that the Larsen C snow pack has a greater amount of liquid water following assimilations. This can be attributed to the fact that the snow pack accumulates water at an earlier stage before the second peak of melting and retains it for a longer period. It should be noted that the values in Figure R1.2 are one order of magnitude higher than those in Table 4 as the quantity of liquid water has been summed over the layers up to a depth of 1 meter, rather than being averaged.

[Figure]

*Figure R1.2 : Comparison of the average cumulated liquid water content of the first meter of the Larsen C snowpack as modeled by MAR without assimilation and the average value of the assimilations.*

*ll. 299-302: this is completely unclear to me.*

> It will be reworded to "*Except for the liquid water content, the snow-related variables (ME, RU, SMB and snowpack density) of the model have undergone significant changes, causing MARref to fall outside the range of the various assimilations. As a result of increased surface melt production, there is an increase in runoff and a subsequent decrease in surface mass balance. This increase in runoff is attributed to the compaction of the upper layers of the snowpack, which reduces its capacity to absorb meltwater.*".

*l. 303: It is not really surprising that the mean of the assimilation experiments is close to the central reference experiment. However without evaluation this is not necessarily meaning that this is more realistic than other members.*

> In this paper, we are not trying to obtain better melt estimates with the assimilations. The main purpose is to test the sensitivity of MAR to the parameters of the assimilation in the aim of evaluating uncertainties on the simulated melt amount. Parts of the articles will be rewritten to state it more clearly. In addition, we want to observe where/when the model is different from the binary mask created with the satellite observations and what are the impacts to match this liquid water extent in

MAR. This point will be discussed in more details and will be more explicit. Here that statement will be reworded to "*In the end, the results illustrate that Assim$_{ref}$ is the closest simulation to $\overline{Assim}$, and makes it an appropriate candidate when computational resources are limited and only the effect of the assimilation want to be studied, not the sensitivity of the model.*"

*Table 4: The caption to this table is sloppily formulated. As in table 3 not all lines and columns are well defined or self explanatory and there is room for guesswork but no sound basis for interpretation. Also I wonder why LWC is consistently one order of magnitude smaller than in Table 3.*

Caption of Table 4 will be rewritten following the new caption of Table 3 and σ will be removed from the table.

In Table 3, we present cumulated values over the whole Peninsula while in Table 4 we present cumulated values only over individual ice shelves. Thus, if we add the values of LWC over the ice shelves and the rest of the studied zone, we will have the values of Table 3. This information will appear clearer in the revised version.

*l. 310: refreezing is indeed releasing energy and heating the ambient snow!*

Yes, refreezing releases energy and heats the snowpack but the quantity of heat released is not sufficient enough to heat up the layers under one meter as even with higer refreeze, Assif$_{ref}$ snowpack temperature under one meter remains similar to MAR$_{ref}$ snowpack temperature. The colder layers will eventually cool down the snowpack.

*Please revise ll. 310-314.*

This part will be reworded to: "*First, the available energy in the system is consumed by melting processes, preventing the layer under 1m from heating up. A colder snowpack constantly needs larger nudging to reach the melt threshold. The second point is that due to the lower saturation of water in the lower layers, the upper layers become saturated with less water because of densification during melt events, resulting in increased runoff and faster percolation of the water into deeper layers, outside of the assimilation depth range. If the model were to retain liquid water in its top snow layers for a longer duration, it would require less nudging to match the RS datasets*"

*l. 310: not sure what prevails means here. Maybe prevail->prevent?*

This sentence will be removed in the revised version as presented in the comment above.

*l. 317: please specify what exactly qualifies results to be improbable. Such an exclusion criterion should be defined beforehand. And the exclusion of the members should be stressed when introducing table 2 in the method section.*

> Exclusion of the simulations will be introduced near line 264 when Table 2 is discussed.
>
> Here, the exclusion criterion was that the Antarctic Peninsula was experiencing negative SMB one order greater than the other simulations when cumulated over the whole melt period on the Antarctic Peninsula. This behavior is not observed in datasets used in other studies (Kittel, 2021; Chuter et al., 2022) and thus may introduce a bias in the comparisons.

*l. 333: lesser -> less or smaller.*

> It will be corrected in the revised version.

*Table 5: Which are the experiments considered here? Are these numbers the same for all experiments with α = 0.1 and α = 0.2 ? Also maybe noteworthy: number of melt days larger for MARref than one of the assimilations on Wilkins.*

> All these experiments are considered. It is the mean values for the assimilation with α = X that is shown. Caption will be changed to "*Comparison between the melt season length and number of melt days modeled for the three studied ice shelves for MAR$_{ref}$ and the average number for assimilations depending on their α for the 2019-2020 melt season.*" to make the message clearer.

*l. 343: is this referring to the whole 20m snow pack?*

> Indeed this important information is missing. Only the first meter of the snowpack is considered, and not the full modeled snowpack.

*l. 347: I don't find these numbers in Table 5- I stop reading this paragraph here.*

> This will be corrected in the revised version as the value given refers to Figure 11. L347-348 will be reworded "*By computing the mean value of each pixel number of melt days of the ice shelves, it was found that the largest deviation occurs on Larsen C, with an increase of 15 melt days. The other two ice shelves exhibit comparatively smaller differences, with Wilkins and Georges VI experiencing an increase of 8 and 9 melt days, respectively.*"

*l. 364: this cycle -> daily melt - refreezing cycle?*

> Yes, the daily melt-refreeze cycle.

*l. 373: is other frequency here higher frequency?*

In this case, the "*other frequency*" should have been "*another frequency than AMSR2*" as it is referring to ASCAT. With a better description of the sensors, there will be less confusion. ASCAT frequency (~5GHz) is lower than AMSR2 (~19GHz).

*Section 5: this probably needs to be rewritten anyway after the other parts of the manuscript have been revised.*

With the new insight on the paper, Section 5 will be rewritten in the revised version. Nonetheless, the conclusion will remain mainly the same. Changes that will be included in section 5 are a more clearer take-home message, a conclusion on the change in liquid water content induced by the assimilation and a small discussion about the evaluation of current surface melt production of the model without assimilation.

*l. 394: effect of assimilation was not studied here.*

"*However, the assimilation of surface melt occurrence has a small impact on the atmosphere.*" will be removed in the revised version.

References:

Chuter, S. J., Zammit-Mangion, A., Rougier, J., Dawson, G., & Bamber, J. L. (2022). Mass evolution of the Antarctic Peninsula over the last 2 decades from a joint Bayesian inversion. *The Cryosphere*, *16*(4), 1349–1367. https://doi.org/10.5194/tc-16-1349-2022

Jakobs, C. L., Reijmer, C. H., Smeets, C. J. P. P., Trusel, L. D., Van De Berg, W. J., Van Den Broeke, M. R., and Van Wessem, J. M. (2020). A benchmark dataset of in situ Antarctic surface melt rates and energy balance. Journal of Glaciology, 66(256), 291–302. https://doi.org/10.1017/jog.2020.6

Kittel, Christoph. (2021). Kittel et al. (2021), The Cryosphere : MAR and ESMs data [Data set]. Zenodo. https://doi.org/10.5281/zenodo.4459259

Kittel, C., Fettweis, X., Picard, G., and Gourmelen, N. (2022a). Assimilation of satellite-derived melt extent increases melt simulated by MAR over the Amundsen sector (West Antarctica), Bulletin de la Société Géographique de Liège, 78, 87–99, 2022.

Kittel, C., Amory, C., Hofer, S., Agosta, C., Jourdain, N. C., Gilbert, E., . . ., Fettweis, X. (2022b). Clouds drive differences in future surface melt over the Antarctic ice shelves. The Cryosphere, 16 (7), 2655–2669. https://doi.org/10.5194/tc-16-2655-2022

---

## Author Comment (AC2)

Reviewer #2

We would like to thank the reviewer #2 for its constructive comments that will improve the paper. We have responded to all of them and will modify our paper accordingly. The majority of the highlighted issues of the paper come from its writing style. The paper will undergo a restructuring and will be sent to a native english speaker after rewording to reduce confusions. Our point-by-point answers (in bleu) follow as a supplement.

Major comments:

*I miss the applicability of this assimilation technique for Antarctica. In Antarctica melt-rates are relatively unimportant in the contemporary climate, and for future simulations this assimilation technique obviously does not work. What will this technique provide us with? Some extra words/paragraphs, either in the introduction, or in the conclusions, or both, should be spent on this to improve the relevance of this paper.*

> In strictly accounting terms, it is true that melting and associated runoff are "second-order magnitude" components of the mass balance computation. However, the surface hydrology triggers other processes (e.g. water loading, ponding, hydrofracturing) that destabilize ice shelves and, therefore, continental ice. In addition, it may not be the case in the future (Gilbert & Kittel, 2021). Unfortunately, we do not have a long time and large-scale observations of the surface melt production in Antarctica that can be used to calibrate models. Remote sensing data are the only dataset for detecting the presence of water we can integrate into the model to constrain the presence of liquid water in the model. The purpose of the paper will be more explicitly integrated into the text of the revised version by concluding the introduction with a paragraph explaining the purpose of the applied technique. The paragraph will be in the style of: "*Assimilation of data into the model is a crucial step in quantifying the uncertainties associated with the model's output without assimilation. The assimilation process helps to identify areas of the model where the modelisations are not consistent with the observations. This can help us to better understand the underlying physical processes and their interactions. In this way, data assimilation provides a powerful tool for improving the reliability of models. In our case, it is an essential step in the process of model refinement, leading to improved predictions of future scenarios.*"

*Slightly related to point 1, I miss a recommendation based on the results of this study. Would the authors advise to use this technique on all future simulations, or is the main aim to provide better uncertainty estimates? I advise the authors to take a stronger stance on what is the main take-home message of the study.*

> As Reviewer 2 mentioned in the first point, remote sensing datasets have only become available in the past few decades, and until now, the Antarctic melt rate may

be considered relatively unimportant. The main aim of the study was to study the sensibility of the model to assimilation to obtain an idea of the uncertainties of the model and intra variability of the assimilation technique itself and modeled melt amount. By obtaining an "ensemble model" that can be compared to MAR, it would be possible to quantify, or at least better estimate, the uncertainties of the model regarding the liquid water content of the snowpack. This would enable the correction of hydrological processes within MAR (without assimilation) to improve its accuracy. This refined mode would enable long-term simulations with a better estimate of the liquid water content of the snowpack in the future. With regards to point 1, a paragraph stating the main message of the paper will be included in the revised version. This paragraph will summarize the relative sensitivity of MAR to the presence of liquid water in the first centimeters of the snowpack and its current predisposition to make water percolate while remote sensing data still observe water.

*I miss a thorough evaluation of the actual modelled surface melt. There are several AWS on the AP that close the SEB (Jakobs, C. L. et al., (2020)) and enable a much more detailed and independent evaluation of simulated melt production. In turn, these can then be used to actually provide (a part of) the uncertainty calculation that the authors hint at in the last sentence of the abstract, which would really improve the papers conclusions and applicability (see point 1).*

As also pinpointed by the other reviewer, the MAR evaluation will be explained in further details in the revised version and the dataset with which the evaluation is performed will be better described. The surface melt can also be evaluated by comparing the results with Jakobs et al. (2020). The results of this evaluation still need to be made carefully. Jakob's dataset also remains a modeled-based estimate, with its own biases and limitations, and therefore cannot be used as if there were in situ measurements, and so a reference for the modeled estimations. However, a short comparison of $MAR_{ref}$ with the AWS used by Jakobs available on The Antarctic Peninsula could be added to the paper.

Nevertheless, it is important to remind that the study does not aim at providing better melt estimates but rather testing the sensitivity of the model. We only evaluate MAR without assimilation as we know that the value given after assimilation may differ from the observations. However, while the MAR simulations without assimilation were conducted for the period of 1980 to 2022, the "assimilations" were only performed from 2019 to 2021, which renders their comparison with the AWS dataset results presented in Jakobs et al. (2020) impossible.

As represented in the Figure here under (Figure R2.1) we compare the melt estimates from the AWS described in Jakobs et al. (2020). to the surface melt

production of the 4 closest MAR pixels of the AWS. MAR has a tendency to overestimate some extremes of melting while simultaneously underestimating or overestimating the duration of periods during which the ice shelves are experiencing melting. Even if it is important to note that there can be a difference in altitude between the AWS and MAR pixels that explains the differences between the two datasets, this comparison also highlight the importance of nudging MAR to correspond to the remote sensing observation of wet snowpack.

[Figure]

*Figure R2.1 : Comparison of surface melt production as modeled by MAR without assimilation and estimated surface melt production from AWS 14, 15, 17, and 18 described in Jakobs et al. (2020).*

Minor (line by line) comments

The paper will be restructured, especially the introduction and methodology, with a stronger contribution of the co-authors and the help of a native English speaker.

*L2-5: unclear. Too much detailed and lengthy information for an abstract, can be considerably shortened by just writing something like the following: "However, RCMS are*

*subject to biases, which Remote Sensing (RS) products can help solving. Here, we assimilate several satellite products that detect surface melt into the RCM MAR…" etc.*

> The abstract will be rewritten to be simplified and focused on essential points. In parallel, other important information such as the problematic related to *in situ* observations and remote sensing will be included in the revised version.

*L10-11: This seems ambigiuous, are the previous two methods not also assimilations?*

> The confusion results from a poor choice of word. The first two parameters are thresholds on assimilation depth and liquid water content used for the assimilation and the third is the choice of assimilated sensors.

*L14-15: Way too detailed for an abstract. Shorten*

> As stated in comment L2-5, the abstract will be shortened and simplified in the revised version. L14-19 will be shortened to only state the sensitivity of MAR to the assimilation depth and its impact on surface melt production.

*L17: A refreeze of what?*

> A (night)refreeze of the meltwater produced during the day. It will be clarified in the text of the revised version.

*L22-23: Good to end the abstract with this (but I expect you to end the conclusions section likewise). Can you extend slightly on this?*

> Conclusion will be extended with a more detailed study of the change in LWC that will be conducted in the result section in the revised version. The study of the evolution of LWC and the saturation of the snowpack will enhance the message of this paper.

*Abstract overall: Please shorten and simplify the abstract!*

*L25: Here you mention both polar ice sheets, and in the following sentence you immediately move to Greenland. This transition can be improved.*

> Sentence will be revised into: "*More than two-thirds of the Earth's freshwater is held in the polar ice sheets (Church et al., 2013), with the majority of it trapped as ice on the ground at the south pole, forming the Antarctic Ice Sheet (AIS). According to Fretwell et al. (2013), if all the ice in the AIS were to melt, it would result in a sea-level rise of 56 meters.*"

*L27: Here you should distinguish between grounded ice mass loss and actual mass loss, especially if later in the introduction you want to emphasize the importance of surface melt (i.e. hydrofracture and grounded ice acceleration)*

The distinction will be made by changing the sentence to: *"Currently, the Antarctic Ice Sheet (AIS) is primarily losing mass due to grounded ice flowing into the ocean. There, the ice is lost mainly through a combination of basal melting and calving."*

*L28: Why is surface melting not yet a big concern now?*

In strictly accounting terms, it is true that melting and associated runoff are "second-order magnitude" components of the mass balance computation. However, the surface hydrology triggers other processes (e.g. water loading, ponding, hydrofracturing) that destabilize ice shelves and, therefore, continental ice. In addition, it may not be the case in the future (Gilbert & Kittel, 2021). Unfortunately, we do not have a long time and large-scale observations of the surface melt production in Antarctica that can be used to calibrate models. Remote sensing data is the only dataset for detecting the presence of water we can integrate into the model to constrain the presence of liquid water in the model. The purpose of the paper will be more explicitly integrated into the text of the revised version by concluding the introduction with a paragraph explaining the purpose of the applied technique. The paragraph will be in the style of: *"Assimilation of data into the model is a crucial step in quantifying the uncertainties associated with the model's output without assimilation. The assimilation process helps to identify areas of the model where the modelisations are not consistent with the observations. This can help us to better understand the underlying physical processes and their interactions. In this way, data assimilation provides a powerful tool for improving the reliability of models. In our case, it is an essential step in the process of model refinement, leading to improved predictions of future scenarios."*

*L36: RCMs are not yet introduced, rephrase*

Noted and it will be changed to *"Regional Climate Models (RCMs) are nowadays one of the effective tools to monitor the ice shelf evolution by enabling to model their past, present, and future climate. For example, MAR (for "Modèle Atmosphérique Régional" in French) has been developed to monitor the polar ice sheets. However, RCMs still have some limitations."* in the revised version.

*L38: what do you mean with "other independent sources of uncertainties". Vague!*

We wanted to state that the data included should not have the same source uncertainty as the model. That is to say that we do not include a dataset that is based on MAR and/or already included in MAR. We will rephrase it *"These*

*uncertainties can be mitigated by employing external data, which is not already indirectly incorporated into MAR, to improve the model's accuracy at specific points in space and/or time."*

*L39-41: How does this assimilation technique compare to other common techniques such as reanalyses?*

Reanalyses such as ERA5 use much more complex assimilation techniques than nudging. Reanalyses assimilate observations in a forecast model by taking into account temporal and spatial variability of the observations. Here, nudging only consists in slightly adjusting the model at each time set to match the observations.

*L42: What do you mean with sequentially? Reword*

Will be rephrased to *"we assimilate satellite-derived surface liquid water presence over the Antarctic Peninsula (AP)"* in the revised version. The world *sequentially* means that, for each time step of the model, we try to match the models and the remote sensing observations. The ontologies related to data assimilation techniques will be explained in subsection 2.3 dedicated to the assimilation algorithm.

*L45: what is a complex surface hydrology?*

Complex in a sense of the variety of the hydrological structure and related processes, and the fact that the water streams do not directly come out of the ice sheet in visible rivers. This part will be reworded to *"These ice shelves undergo most of the surface melt of the AIS. Their surface and subsurface processes are poorly understood due to challenges in making direct observations caused by their complex surface hydrology. (Barrand et al., 2013; Datta et al., 2019; Johnson et al., 2020). "*

*L46: isn't it more like 10km scale?*

It depends on the size of the model domain. The model could have been run at 5 km instead of 7.5 km. It will be changed to "10 km scale" to be more realistic with the diversity of spatial resolution used in literature.

*L47: rephrase.*

It will be rephrased to *"Phenomena such as melt induced by the Foehn effect can occur at a smaller spatial scale than the spatial resolution of RCMs and thus may not be correctly represented the model (Datta et al., 2019; Chuter et al., 2022; Wille et al., 2022)."*

*L49: "multiple" comes out of the blue and confuses me, rephrase.*

It will be rephrased to "I*n this study, we use four remote sensing datasets. This enables us to perform assimilation over the entire studied zone every day for two melt seasons (2019-2020 and 2020-2021), even if one of the datasets is missing acquisitions for one or multiple days.*" in the revised version.

*L50-51: Is it, or will it be, a promising technique? Outside of Kittel 2022, there is not really any other study doing this right?*

Assimilation in general is a promising technique, especially with the increasing amount of spaceborne sensors, as well as the longer and longer Earth Observation time series available. Multiple studies are already comparing models and remote sensing data. But assimilating remote sensing data in models is not new, Navari et al. (2016,2018) are examples of *posterior* data assimilation in MAR. But in the case of assimilation of wet snow retrieval into MAR with nuddging, my current knowledge only encompasses the work of Kittel et al. (2022).

*L52: Again, vague, and repeat of the previous.*

This line will be removed in the revised version.

*Figure 1: It's George VI, not Georges.*

Noted and it will be changed accordingly in the revised version. Same for other mentions of George VI in the text.

*L54-55: you already mentioned this in the beginning of the introduction.*
*This paragraph needs to be rewritten or completely removed; most info is repeated or obsolete. Your paper is about assimilating melt, so spend time on explaining melt and why it is important to improve melt simulations.*

With the restructuring of the introduction, the paragraph will be moved and better integrated at the beginning of the introduction to be merged with the line where this is already stated.

*L67-72: This paragraph is all over the place, again repeating previously introduced information. Rephrase it and make it more concise by just writing: "Here, we assimilate different satellite observations of melt in the RCM MAR, etc". The Methods section is to explain the actual details, pros and cons of the products.*

These lines are supposed to explain the main advantages/disadvantages of the passive/active sensor and that both can be used together to benefit from the advantages of the two types of sensors. We still believe this paragraph contains important information as a remote sensing point-of-a-view, but requires a complete

rebuild. Section 2 will include subsections describing the remote sensing products where the pros and cons will be explained.

*L78: Introduce what a binary melt mask is, I did not know.*

"*Binary melt masks*" is an unwise choice of word that means we created masks of 1 and 0 from the satellite observations. 1 signifies wet snow will be detected and 0 the negative. "*Binary melt masks*" will be reworded to "*wet snow masks*" in the revised version and will be introduced in L78.

*L78: Three sensors? Sensors on the satellites? The three satellites? Unclear.*

It is true that the remote sensing part may be confusing. The spaceborne carriers and their sensors are often interchanged, making the information difficult to follow. Subsections will be created for each remote sensing dataset employed during the study. Subsections will include a description of the mission, sensor, and its characteristics. Better describing the datasets should decrease the confusion around them.

*L79: radiometer is a new word, it should be introduced. Are all satellites equipped with microwave radiometers?*

A radiometer is a sensor for measuring the radiant flux of electromagnetic radiation. Here only AMSR2 is a radiometer. The other two are active sensors, which means that they do not only observe the radiant flux, they send energy as pulsed electromagnetic waves to the Earth and record the backscattered portion of the signal.

*L84: this is not correct. Liquid water can't be melting. Rephrase to something like: "Here, we relate subsurface liquid water with subsurface melting". Although I am still confused how this works, how do you distinguish between percolated surface meltwater and subsurface meltwater, or meltwater that has not yet refrozen after a previous melt event?*

In the paper, the presence of liquid water observed in the remote sensing dataset but not in the model is considered meltwater. The wording was chosen because we force melt in the model to match the observed presence of liquid water in the snowpack from the satellites. However, this wording is confusing as we do not observe melting with satellites but the presence of water. We will rework the wording by changing the ontology:

- "*observed melt*" → "*observed wet snow*"
- "*binary melt masks*" → "*wet-snow masks*",
- "*assimilated melt state*" → "*assimilated liquid water content state*",

and so on.

*L85: I am unfamiliar with remote sensing so have no idea what you mean by acquisition capabilities.*

Here we stated that by using sensors operating in microwave frequencies, it is possible to obtain images even at night or when there are clouds. This confusion can be clear out by mentioning what "*capability*" we're refering to, e.g. *"day-and-night capabilities"*.

*L86: specify "small scale".*

Small scale is used to talk about melt events with an extent under 100km² which is smaller than the spatial resolution of radiometers.

*L97: Rephrase to "A dry snowpack has a lower emissivity than a wet snowpack".*

It will be rephrased accordingly in the revised version.

*Equation 1: Is it epsilon^*, or is * a multiplicator? Anyhow epsilon is not defined in the text.*

It is a multiplicator. Epsilon is defined at L97 as the emissivity of the snowpack. It will be stated under the equation as well in more general terms.

*L108: I don't understand, what's "dominant melt"? rephrase*

It will be rephrased to "*The grids are superimposed, and the melting state for each pixel in the MAR is determined based on the most prevalent melting or non-melting condition observed in the corresponding area of the satellite mask.*"

*L111: Please group the three satellite production per subsection.*

Subsections will be created in the revised version for each remote sensing dataset employed during the study. Subsections will include a brief description of the mission, the sensor, and its main characteristics. Better describing the datasets should decrease the confusion around them.

*L143: "pixel-wise"? huh? Typo?*

"*pixel-wise multiplication*" refers to a mathematical operation where each pixel in one image or gridded data set is multiplied with the corresponding pixel in another image or gridded data set. In algebra, the corresponding operation is called the Hadamard product. Pixel-wise can also be used in other contexts such as pixel-wise comparison, pixel-wise classification, pixel-wise cross entropy, as examples.

*Figures 2-4: Is there a way to combine these in one graph, or something? They seem rather obsolete to this study (using 3 figures to show something that's not main result of the paper).*

> Comments from anonymous reviewers #1 and #2 tend to suggest opposite directions (either simplify or on the contrary go into more detail). Consequently, the figures may change. Considering this viewpoint, having 3 figures (Fig. 2,Fig. 3, and Fig. 4) on melt detection, which is not the main subject of this paper, may be redundant. Fig 3 will be removed as well as its reference line 104. Also, Figure 4 will be reworked in order to add a panel showing the b) panel after normalization.

*Section 2.3.1: So, what threshold do you finally choose? This is unclear.*

> Both 0.1 and 0.2 thresholds are used in the different assimilations. It will be added to the end of the subsection. They are also mentioned during naming conventions for the different assimilations. We plan to add *"Currently, there is no clue to identify the best fitting threshold for this study. Both thresholds will be used to test the sensibility of the model."*

*L235: It's unclear for me what you are presenting here. Are you evaluating your melt assimilation simulations, or are you repeating evaluations from previous studies? It the latter, this entire paragraph is obsolete.*

> The evaluation of MAR prior to its assimilation is a mandatory step to optimize its hyper-parametrization. From this benchmarked model, we can then add the assimilation module whose sensitivity is the object of the paper. The parametrization used here is based on Kittel (2021), but over a different region, resolution, and time period, thus needs to be assessed. Section 3 thus presents $MAR_{ref}$ evaluation.

*Section 3: The evaluation should be more detailed. AWS observations exist that are used in a SEB model so explicitly calculate melt. This can be perfectly used to evaluate the model, especially the later sensitivity experiments, and assess the models performance in simulating surface melt production. See for instance Jakobs et al., 2020.*

> See our earlier comment about this in major comment number 3.

*L280: rephrase.*

> l279 - 281 will be reworded to *"Despite the fact that the relative increase in surface melt and runoff is almost similar for $\overline{Assim}$ and $MAR_{ref}$ (66.7% and 63.8%, respectively), their absolute increase in Gt y-1 is not the same (+95 Gt y−1 and +21 Gt y−1, respectively). This suggests that the snowpack can still absorb liquid water unless it reaches its maximum capacity."*

*Results overall: Several sentences in the results section are better suited in the introduction or methods section; please increase the focus of this section on the actual results.*

As the introduction and methodology section will be reworked, some sentences from the results section will be moved there, such as the presentation of the different simulations.

*L295: What's the global zone? And what evolution?*

"*Global zone*" refers to the whole Antarctic Peninsula and the "*evolution*" to the change of the variable with and without assimilation. This will be reworded to "*For the three highlighted ice shelves (Larsen C, Wilkins, and Georges VI), the effect of the assimilation follows the same general trend as for the whole peninsula but at different orders of magnitude (Table 4).*" in the revised version.

*Section 4.1: This section is completely unclear to me and contains several unphysical explanations (e.g. a cold snowpack producing melt??). And, I don't understand Figure 10. What are the curves? Not all curves are explained in the legend and as most of them overlap I also can't distinguish them at all. Improve the figure and try to extend the caption.*

Section 4.1 will include a discussion about the comparison between the evolution of melt production, refreeze, and liquid water content that should clarify section 4.1. L310 - 314 will be reworded to: "*First, the available energy in the system is consumed by melting processes, preventing the layer under 1m from heating up. A colder snowpack constantly needs larger nudging to reach the melt threshold. The second point is that due to the lower saturation of water in the lower layers, the upper layers become saturated with less water because of densification during melt events, resulting in increased runoff and faster percolation of the water into deeper layers, outside of the assimilation depth range. If the model were to retain liquid water in its top snow layers for a longer duration, it would require less nudging to match the RS datasets.*"

As for Figure 10, its purpose is to show that curves of the same colors (same assimilation depth for AMSR2 data but different assimilation depth for Sentinel-1 data) have mainly the same outputs. This figure will be reworked to make it clearer in the revised version.

*L306: the penetration depth of what? Be a bit more explicit.*

It will be reworded to "*Assimilation depth*".

*L311-313: Rephrase. What do you mean here?*

The end of the paragraph will be reworded.

*L337-338: How is that calculated? As the melt season starts in November of the previous year?*

> "*Melt season length*" is here defined as the number of days between the first and last day where liquid water is observed in the snowpack during the period 01/06/YYYY to 31/05/YYYY+1.

*L355-368: I think this can all be moved to the Methods section (and in fact, it already contains several things already introduced in the methods).*

> The beginning of section 4.3 will be shortened and simplified as most of its content is already stated before. L335-336 will be removed as it is a repetition of information already stated.

*L372: is this the sum of three ice shelves, or the whole AP? This is unclear. Also elsewhere in the next paragraphs. Be very consistent with these numbers.*

> The numbers are referring to the whole Antarctic Peninsula. "*The two assimilations gave similar numbers of melt days and close surface melt production on the Peninsula ice shelves*" reworded to "*The two assimilations gave similar numbers of melt days and surface melt production on the Antarctic Peninsula for the studied period*"

*L388: uncomplete sentence.*

> It will be reworded to: "*We identified the assimilation depth (Δz) to be the most influential parameter when applied for low penetrating sensors.*"

*L390-391: rewrite.*

> L390-391 will be reworded to "*The uppermost layer of the snowpack is considerably denser than the underlying layers, owing to refreezing caused by the exceeding liquid meltwater from assimilation, as well as low night-time temperatures*" in the revised version.

References:

Jakobs, C. L., Reijmer, C. H., Smeets, C. J. P. P., Trusel, L. D., Van De Berg, W. J., Van Den Broeke, M. R., and Van Wessem, J. M. (2020). A benchmark dataset of in situ Antarctic surface melt rates and energy balance. Journal of Glaciology, 66(256), 291–302. doi:10.1017/jog.2020.6.

Kittel, C., Fettweis, X., Picard, G., and Gourmelen, N. (2022a). Assimilation of satellite-derived melt extent increases melt simulated by MAR over the Amundsen sector (West Antarctica), Bulletin de la Société Géographique de Liège, 78, 87–99, 2022.

Kittel, C., Amory, C., Hofer, S., Agosta, C., Jourdain, N. C., Gilbert, E., . . ., Fettweis, X. (2022b). Clouds drive differences in future surface melt over the Antarctic ice shelves. The Cryosphere, 16 (7), 2655–2669. https://doi.org/10.5194/tc-16-2655-2022

Navari, M., Margulis, S. A., Bateni, S. M., Tedesco, M., Alexander, P., & Fettweis, X. (2016). Feasibility of improving a priori regional climate model estimates of greenland ice sheet surface mass loss through assimilation of measured ice surface temperatures. The Cryosphere, 10 (1), 103–120. doi: 10.5194/tc-10-103-2016.

Navari, M., Margulis, S. A., Tedesco, M., Fettweis, X., & Alexander, P. M. (2018). Improving greenland surface mass balance estimates through the assimilation of modis albedo: A case study along the k-transect. Geophysical Research Letters, 45 (13), 6549-6556. doi: 10.1029/2018GL078448.

---

## Referee Report (RR1)

Review of *The Cryosphere* Manuscript egusphere-2022-1371:

"Sensitivity of the Regional Climate Model MAR's snowpack to the assimilation parametrization of satellite-derived wet-snow masks on the Antarctic Peninsula"

Authors: Dethinne *et al.*

Received: July 17, 2023
Reviewed: July 28, 2023

Reviewer: Charlie Zender, `zender@uci.edu`

Recommendation: Accept subject to minor revisions

I have voluntarily disclosed my identity in all manuscript reviews since 2004. The authors are free to contact me at `zender@uci.edu`.

**General Comments**

This manuscript reports on the sensitivity of the MAR model to different assimilation methods for satellite-observations of wet snow over the Antarctic Peninsula. The manuscript summarizes a fairly comprehensive battery of numerical experiments to help identify the strengths and weaknesses of various assimilation datasets and methods. It will be of interest to *The Cryosphere* readers who are interested in understanding and reducing the uncertainties and biases of a cutting-edge Regional Climate Models (RCMs) for polar regions.

I was invited to weigh-in on the suitability of the revised (not the original) manuscript for publication. Reviewers 1 and 2 delivered substantial comments on technical as well as overarching goals of the original manuscript. The authors responded with major revisions that attempted to address all of many points raised. The authors characterize the reviews this way: "The majority of the highlighted issues of the paper come from its writing style." This may be true in terms of the number of comments, yet the reviews contain many substantial and trenchant scientific concerns.

That said, the authors have adequately addressed the vast majority of the reviewers comments of all kinds. In particular, the addition of tables, adoption of consistent and systematic naming conventions, and clarification of manuscript intent greatly clarified and focused the manuscript's content. The revised manuscript is of sufficiently high quality to warrant publication after addressing a few more scientific points, and another thorough edit to address the English syntax/spelling issues.

**Specific Comments**

1. The methodology used to assimilate the RS data is impressive. Nice work!

2. L206: "as very little [sic] melting events are expected". Winter-season melt is not uncommon on Larsen C during Foehn events. More than 20% of annual melt at AWS18 in Cabinet Inlet on Larsen C has occurred due to sensible heating by foehns during polar night (*Kuipers Munneke et al.*, 2018). *Laffin et al.* (2021) and *Laffin et al.* (2022) quantify seasonal and regional behavior of such foehn-induced melt on the eastern AP.

3. Lines 231–239 repeat lines 221–229 verbatim. Oops :)

4. Large portions of lines 280–292 repeat lines 272–279 verbatim.

5. Where is the exact region of the Antarctic Peninsula (AP) simulated by MAR defined? Tables 4 and 5 and Figures 7–9 appear to tabulate the entire AP, and then some. Where is the region for which the mass budgets are computed defined? The present results cannot be reproduced without a clear definition of the boundaries.

6. Line 355 defines the accumulation portion of SMB as comprising snowfall and rainfall. However, the ablation portion only mentions runoff and sublimation, not evaporation. This seems inconsistent, as evaporation from the liquid phase does reduce SMB during melt (or rainfall) events just as sublimation reduces SMB during dry weather.

7. Line 355: In accord with Reviewer 1's suggestion, the revised manuscript should explicitly state that MAR is configured not to represent snow drift, so it is not included in the SMB.

8. The relative roles of the three assimilation parameters studied in influencing densification, runoff, and SMB is interesting. The manuscript reaches conclusions that will be helpful in advancing assimilation methods and reducing biases in RCMs.

9. The maximum volumetric liquid water content of firn prior to percolation adopted by MAR and used in this study is 5%. Where does the experimental support for this limit described? Please comment on the expected sensitivity the assimilation to the value chosen for this parameter within its uncertainty range. Please cite or describe the original source/justification for this limit the first time it appears in the manuscript.

**Technical Corrections**

The revised manuscript contains fewer, though still many, instances of poor English syntax, verb use, spelling, and adjective placement. A fluent English speaker could catch those remaining instances. On the whole, though, I understood the intended meaning of virtually the entire manuscript, including the awkwardly written sentences. The following list is very incomplete, and meant to be illustrative rather than comprehensive of the editing task that remains:

1. L29: "thought" not "taught"
2. L55: "RCMs" not "RMCs"
3. L62: "quantify the surface meltwater quantity" is redundant
4. L83: Replace "one is called" by "AMSR2 is"

5. L83: "Platform" not "Plateform"
6. Figure 2 caption: "Temperature brightness (K) the 09-06-2019" is bad English
7. Figure 3 caption: should "between (a) and (b)" be "between (b) and (c)"
8. L271: "2.3.3 Experiments conducted" not "2.3.3 Experiences conducted"
9. L308: "are punctual when"?
10. L347: "almost similar" is almost redundant, just say "similar"

**References**

Kuipers Munneke, P., et al. (2018), Intense winter surface melt on an Antarctic ice shelf, *Geophys. Res. Lett.*, *45*(15), 155–165, doi:10.1029/2018GL077899. 2

Laffin, M. K., C. S. Zender, S. Singh, J. M. van Wessem, C. J. P. P. Smeets, and C. H. Reijmer (2021), Climatology and evolution of the Antarctic Peninsula föhn wind-induced melt regime from 1979–2018, *J. Geophys. Res. Atm.*, *126*(2), doi:10.1029/2020JD033682. 2

Laffin, M. K., C. S. Zender, M. van Wessem, and S. Marinsek (2022), The role of föhn winds in eastern antarctic peninsula rapid ice shelf collapse, *The Cryosphere*, doi:10.5194/tc-16-1369-2022. 2

---

## Author Response (AR2)

Reviewer : Charlie Zender

We would like to thank the reviewer for his constructive comments that will improve the paper. We have responded to all of them and modified our paper accordingly. The reviewer was invited to judge the revised paper, not the original one. The reviewer explains that after the major revision, the paper has been greatly clarified and focused but a few more scientific and writing point still remains to be addressed. Our point-by-point answers (in blue) follow as a supplement.

Specific comments:

1. *The methodology used to assimilate the RS data is impressive. Nice work!*
   Thank you, the methodology has been studied to maximize the utilization of remote sensing data while avoiding over-reliance on them for the model.

2. *L206: "as very little [sic] melting events are expected". Winter-season melt is not uncommon on Larsen C during Foehn events. More than 20% of annual melt at AWS18 in Cabinet Inlet on Larsen C has occurred due to sensible heating by foehns during polar night (Kuipers Munneke et al., 2018). Laffin et al. (2021) and Laffin et al. (2022) quantify seasonal and regional behavior of such foehn-induced melt on the eastern AP.*
   Indeed melt events can occur during winter on Larsen C during winter. The sentence in line 206 is too brief to provide a complete explanation of why assimilation is not carried out during winter. The complete reason is based on two points.
   First, there is a computational reason. The assimilation increases the computation time of MAR by a factor of 2 when the routine is on. As the Assimilation was planned to be performed over the entire Antarctic ice sheet and for a longer period, we disable it during winter as fewer melt events would occur on average (Figure 6).
   Second, it would be very interesting to study those melt events during winter. But as they are caused by specific events (mostly foehn-induced melt), we choose not to include them in the study of the general sensitivity of the model to the parameter of the assimilation.
   To clarify the situation, the sentence will be rewritten to: "*Outside of this timeframe, no assimilation is performed for computational constraints and the likely prevalence of shorter melting events during winter. These events are commonly related to Foehn effects near the grounding line (Munneke et al., 2018) where the effectiveness of passive sensors decreases*".

3. *Lines 231–239 repeat lines 221–229 verbatim. Oops :)*

We are deeply sorry for this mistake. Lines 221-229 have been removed in the revised version of the paper. An error in the LaTeX code includes the lines in the text instead of including them in the track change.

4. *Large portions of lines 280–292 repeat lines 272–279 verbatim.*
   Same as for comment number 3, we are deeply sorry for this mistake. Lines 272-279 have been removed in the revised version of the paper. An error in the LaTeX code includes the lines in the text instead of including them in the track change.

5. *Where is the exact region of the Antarctic Peninsula (AP) simulated by MAR defined? Tables 4 and 5 and Figures 7–9 appear to tabulate the entire AP, and then some. Where is the region for which the mass budgets are computed defined? The present results cannot be reproduced without a clear definition of the boundaries.*
   The exact MAR domain is a bit larger than Figure 1. After removing the edges of the domain, its extent was reduced to the same extent as in Figure 1. For the ice shelves, the extent used is the same as the extent highlighted in Figure 1. The revised version of the paper will include :
   - In line 191: *"The spatial extent of the simulations corresponds to the extent of Figure 1."*
   - The caption of Figure 1 will be updated with *"The red square around the AP corresponds to the MAR spatial extent."*
   - Captions of Tables 4 and 5 will be updated with *"[...] over the 2019-2020 melt season for the whole MAR spatial extent."*
   - The caption of Figures 7-8-10-11-13 will be updated with *"[...] over the whole MAR domain for the 2019-2020 melt season as modeled by MAR [...]"*
   - Captions of Tables 6 and 7 will be updated with *"[...] over the three highlighted ice shelves in 2019-2020 using the delimitation shown in Figure 1."*

6. *Line 355 defines the accumulation portion of SMB as comprising snowfall and rainfall. However, the ablation portion only mentions runoff and sublimation, not evaporation. This seems inconsistent, as evaporation from the liquid phase does reduce SMB during melt (or rainfall) events just as sublimation reduces SMB during dry weather.*
   Here, "sublimation" is used to refer to all water fluxes, i.e. the condensation, evaporation, and sublimation balance. Evaporation will be added in the ablation term in line 356.

7. *Line 355: In accord with Reviewer 1's suggestion, the revised manuscript should explicitly state that MAR is configured not to represent snow drift, so it is not included in the SMB.*

The received version of the manuscript will include at the end of line 190: "The blowing snow module of MAR is not used in this study causing no representation for snow drift.".

8. *The relative roles of the three assimilation parameters studied in influencing densification, runoff, and SMB is interesting. The manuscript reaches conclusions that will be helpful in advancing assimilation methods and reducing biases in RCMs.*

Thank you. Your recognition of the importance of our study is deeply encouraging and truly motivating.

9. *The maximum volumetric liquid water content of firn prior to percolation adopted by MAR and used in this study is 5%. Where does the experimental support for this limit described? Please comment on the expected sensitivity the assimilation to the value chosen for this parameter within its uncertainty range. Please cite or describe the original source/justification for this limit the first time it appears in the manuscript.*

Regarding the maximum volumetric liquid water content of firn prior to percolation as adopted by MAR and utilized in our study, we acknowledge the valid question raised. While the order of magnitude of the irreducible water saturation in MAR is based on Coléou & Lesaffre (1998), the specific choice for this limit is not currently described in existing publications (at least, for simulation in Antarctica (Reijmer et al., 2012)), the choice of this value is based on considerations that align with the MAR model's development context. Regarding assimilation, this parameter can indeed impact the outcomes, yet no sensitivity analyses were conducted. By retaining increased liquid water content in the initial layers, it could potentially necessitate reduced nudging to align with RS datasets during extended periods of wet snow without inducing excessive melting but it may induce more refreeze as explained in Fettweis et al. (2011).

In a matter of clarity, we will reword lines 186-188 to: "*Each layer has a maximum liquid water content (LWC) of 5 % of its air content beyond which the water freely percolates to the deeper layer or runoffs above impermeable layers (bare ice or ice lenses) (Coléou & Lesaffre, 1998).*" in the revised version of the manuscript. We will also discuss the effect of this choice by rewording lines 398-399 to "*If the model were to retain liquid water in its top snow layers for a longer duration, it would require less nudging to match the RS datasets. Achieving this could involve increasing the maximum liquid water content of the snow layers. However, enhancing water retention in the initial snowpack layers might lead to increased refreezing and*

*consequently, densification, depending on the snowpack temperature (Fettweis et al. 2011)."*

Technical Corrections:

*The revised manuscript contains fewer, though still many, instances of poor English syntax, verb use, spelling, and adjective placement. A fluent English speaker could catch those remaining instances.*

We acknowledge that the paper still contains poorly written sentences. The paper is currently undergoing revisions by a native English speaker. Among other minor syntax and grammar corrections, technical corrections 1 to 10 have been duly incorporated in the revised version.

References:

Coléou, C. & Lesaffre, B. (1998). Irreducible water saturation in snow: Experimental results in a cold laboratory. Annals of Glaciology. 26. https://doi.org/10.1017/S0260305500014579

Reijmer, C. H., van den Broeke, M. R., Fettweis, X., Ettema, J., and Stap, L. B. (2012). Refreezing on the Greenland ice sheet: a comparison of parameterizations. The Cryosphere, 6, 743–762, https://doi.org/10.5194/tc-6-743-2012

Fettweis, X., Tedesco, M., van den Broeke, M., and Ettema, J. (2011). Melting trends over the Greenland ice sheet (1958–2009) from spaceborne microwave data and regional climate models. The Cryosphere, 5, 359–375, https://doi.org/10.5194/tc-5-359-2011

Munneke, P. K., Luckman, A. J., Bevan, S. L., Smeets, C. J. P. P., Gilbert, E., Van Den Broeke, M. R., Wang, W., Zender, C., Hubbard, B., Ashmore, D., Orr, A., King, J. C., & Kulessa, B. (2018). Intense winter surface melt on an Antarctic ice shelf. Geophysical Research Letters, 45(15), 7615-7623. https://doi.org/10.1029/2018GL077899